# Aligning Compound AI Systems via System-level DPO

**Xiangwen Wang** [1,2*]    **Yibo Jacky Zhang**[1*]    **Zhoujie Ding**[1]

**Katherine Tsai**[1]    **Haolun Wu**[1,3]    **Sanmi Koyejo**[1]

[1]Stanford University    [2]University of Science and Technology of China    [3]Mila - Quebec AI Institute
xiangwen@stanford.edu, yiboz@stanford.edu, d1ng@stanford.edu
tsaikl@stanford.edu, haolunwu@cs.stanford.edu, sanmi@cs.stanford.edu

## Abstract

Compound AI systems, comprising multiple interacting components such as LLMs, foundation models, and external tools, have demonstrated remarkable improvements compared to single models in various tasks. To ensure their effective deployment in real-world applications, aligning these systems with human preferences is crucial. However, aligning the compound system via policy optimization, unlike the alignment of a single model, is challenging for two main reasons: (i) non-differentiable interactions between components make end-to-end gradient-based optimization methods inapplicable, and (ii) system-level preferences cannot be directly transformed into component-level preferences. To address these challenges, we first formulate compound AI systems as Directed Acyclic Graphs (DAGs), explicitly modeling both component interactions and the associated data flows. Building on this formulation, we introduce **SysDPO**, a framework that extends Direct Preference Optimization (DPO) to enable joint system-level alignment. We propose two variants, **SysDPO-Direct** and **SysDPO-Sampling**, tailored for scenarios depending on whether we construct a system-specific preference dataset. We empirically demonstrate the effectiveness of our approach across two applications: the joint alignment of a language model and a diffusion model, and the joint alignment of an LLM collaboration system.

## 1 Introduction

Compound AI systems, which consist of multiple interacting AI components, serve as promising frameworks to push beyond model capabilities and achieve state-of-the-art performance [40, 6, 13, 16]. For example, ChatGPT integrates a Large Language Model (LLM), a DALL-E image generator, a web browser plugin, and various other system components to support diverse user needs [1]. A multi-agent system consisting of multiple LLMs working collaboratively achieves improved performance compared to a single agent [36]. A Retrieval-Augmented Generation (RAG) system combines large language models with information retrieval capabilities and is capable of answering time-sensitive queries. A multi-LLM routing system includes a router that dynamically selects among a diverse set of models according to user queries to maximize model performance [11]. These examples illustrate how compound AI systems leverage LLMs alongside complementary modules to tackle complex tasks beyond the reach of a single LLM.

Ensuring effective collaboration among components is crucial for compound AI systems to function reliably. It also plays a critical role in aligning the outputs of the system with human preferences and in ensuring safety and ethical standards [16]. However, simply integrating multiple models does not guarantee effective coordination, as illustrated by a failure case involving an LLM (GPT-4) and a

---

*Equal contribution

39th Conference on Neural Information Processing Systems (NeurIPS 2025).

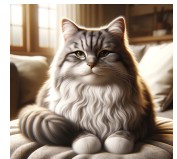 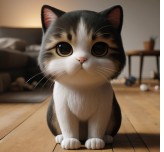 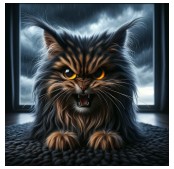 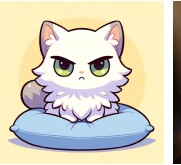 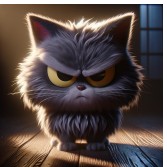 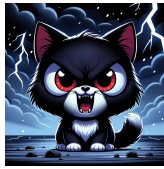

| Calm Cat | Slightly Irritated Cat | Very Angry Cat | Slightly Annoyed Cat | Angry Cat | Furious Cat |

(a)                        (b)

Figure 1: The figure illustrates the challenges in a compound system composed of the GPT-4 and the image generator DALL-E. Given the user prompt to GPT-4, "`Generate three separate images of a cat being progressively angrier`", the task is to demonstrate *a clear visual progression of the specified attribute, i.e., anger*. **(a)** shows the results from one query, and **(b)** represents the results from another query. The captions under each image summarize the prompts generated by GPT-4 for DALL-E (complete prompts in Appendix A), where prompts from both queries reflect progressions in anger. Similarly, DALL-E accurately generates the images following the given prompts. However, **(a)** fails to demonstrate a clear visual progression of anger compared to **(b)**, highlighting GPT-4's inconsistent collaboration with DALL-E.

diffusion model (DALL·E), shown in Figure 1. Our experiments (Section 5) reveal that an instructed tuned Llama-3-8B combined with Stable Diffusion XL achieve a correctness rate of only 32% on similar tasks. These failure cases highlight the critical need to develop a new framework to align compound AI systems.

While alignment techniques for monolithic models are well-studied [26, 44, 3], aligning compound systems remains an open challenge. Standard methods such as Direct Preference Optimization (DPO) [26] and Reinforcement Learning from Human Feedback (RLHF) [19, 44, 3] are not directly applicable to compound systems for three primary bottlenecks: *(i) Non-differentiable interactions*: Components in a compound system often interact through non-differentiable channels, such as natural language, preventing end-to-end optimization and making credit assignment across components difficult. *(ii) Non-decomposable preferences:* Aligning each component independently is inadequate, as system-level preferences are not simply decomposable into individual preferences. Moreover, effective coordination between components is essential but cannot be captured through isolated alignment. *(iii) Lack of fine-grained benchmarks:* Most alignment benchmarks are constructed to evaluate the entire system; benchmarks for individual sub-tasks might not exist.

In light of these challenges, there is an urgent need to develop methodologies for aligning compound AI systems. While recent studies investigate prompting techniques and instruction tuning approaches [39, 16, 30], and concurrent work [37] explores optimization using local reward models, we take an alternative approach. Our main contributions are summarized below:

- We model compound AI systems as Directed Acyclic Graphs and propose *SysDPO*, a Direct Preference Optimization (DPO)-based alignment framework with two variants—*SysDPO-Direct* and *SysDPO-Sampling*—for settings with or without system-specific datasets (Section 2).

- We provide a theoretical analysis showing that *SysDPO* achieves $\beta$-perfect alignment in the population setting, generalizing standard DPO guarantees to compound systems (Section 3).

- We demonstrate *SysDPO* with two applications: aligning an LLM and a text-to-image diffusion model, as well as aligning two LLMs (Section 4). Our experimental results indicate that aligning compound AI systems increases the success rate in handling complex instructions (Section 5).

These results deepen our understanding of alignment challenges in compound AI systems and provide a foundation for future research.

## 2 The SysDPO Framework

In this section, we introduce the framework of SysDPO through intuitive motivations, where the theoretical justification is provided later in Section 3. We start by reviewing prior work on the Bradley-Terry model [4] and DPO [26], then move to introduce the SysDPO pipeline.

**Bradley-Terry (BT) model.** Given an input $x$, the system generates two pairs of outputs $z, z'$. We represent the preference of the outputs as $(z \succ z' \mid x)$ if $z$ is preferred over $z'$ by a preference oracle, e.g., human labelers. To model the preference, we use the Bradley-Terry model – a common

preference model used in alignment [26, 31, 3], which represents the preference distribution as

$$\text{pref}(z \succ z' \mid x) = \frac{\exp(r^*(x, z))}{\exp(r^*(x, z)) + \exp(r^*(x, z'))} \in (0, 1), \tag{1}$$

where $r^*(x, z)$ is the ground truth reward model. Drawing preference from the preference oracle, the winning sample is assigned to $z^w \leftarrow z$ and the losing sample is assigned to $z^l \leftarrow z'$ with probability $\text{pref}(z \succ z' \mid x)$. Using the preference oracle, one can construct a preference distribution $\mathcal{D}$ composed of preference pairs $(x, z^w, z^l)$.

**Direct Preference Optimization.**   DPO [26] aligns the model $\theta$ using the preference distribution $\mathcal{D}$ by minimizing the following loss:

$$L(\theta) = -\mathbb{E}_{(x, z^w, z^l) \sim \mathcal{D}} \left[ \log \sigma \left( \beta \log \frac{p_\theta(z^w|x)}{p_{\bar\theta}(z^w|x)} - \beta \log \frac{p_\theta(z^l|x)}{p_{\bar\theta}(z^l|x)} \right) \right], \tag{2}$$

where $\bar\theta$ denotes the reference model of $\theta$, $\sigma(\cdot)$ stands for the sigmoid function. When it comes to compound AI systems, the optimization challenge arises from the structure of $\theta$, which represents a collection of model parameters. These models may communicate in complex ways that are non-differentiable, e.g., by exchanging plain text or task-specific outputs. Moreover, for a compound system with intermediate generations $y$, the system's generation probability takes the form of an integral $p_\theta(z \mid x) = \int p_\theta(z, y \mid x) \, \mathrm{d}y$, raising challenges for optimization. Therefore, the alignment of compound AI systems is an important yet difficult task.

## 2.1   SysDPO Framework

To circumvent the challenge of non-differentiability, we develop SysDPO. We start by modeling the structure of compound AI systems as DAGs, which encode both the connections between models and the underlying data flow from input to final output via intermediate results. The DAG structure enables us to decompose the joint probability of generated outputs into several components. We study two decomposition methods, depending on whether the intermediate outputs are observable or not. Such decompositions lead to two variations of SysDPO: *SysDPO-Direct* and *SysDPO-Sampling*. Both methods address the non-differentiability and optimization issues. We then define a DPO-based loss function that can be optimized from end-to-end simply via gradient descent. This ensures that the outputs of each component is aligned with human preferences.

**Formulating Compound AI Systems as DAGs.**   We model a compound AI system as a Directed Acyclic Graph (DAG), where nodes represent variables and edges capture the flow of information between components. Specifically, we define nodes as $x$, $\{y_i\}_{i \in I}$, and $\{z_j\}_{j \in J}$, where $x \in \mathcal{X}$ is the input, $y_i \in \mathcal{Y}_i$ are intermediate outputs, and $z_j \in \mathcal{Z}_j$ are final outputs. Each non-input node is generated by a single model that consumes inputs from its parent nodes. We denote the set of all generated outputs by $s = \{y_i, z_j\}_{i \in I, j \in J}$. The directed edges represent the flow of the generated data between components.

We illustrate this formulation with two examples. The first, shown in Figure 2 (a), involves an LLM generating image captions, followed by a diffusion model that synthesizes images—corresponding to the motivating example in Figure 1. The second example, shown in Figure 2 (b), consists of two LLMs collaborating in a multi-stage pipeline. This setup reflects recent interest in LLM collaboration in improving reasoning, factuality, safety, and creativity through mechanisms such as verification, debate, or response refinement [42, 8, 7, 36, 38].

## 2.2   SysDPO-Direct

The DAG structure encodes the conditional independence of the generated data [20], allowing the decomposition of the probability of generated data into multiple terms. Assume that all intermediate outputs $\{y_i\}_{i \in I}$ are observed or given in the preference dataset. We can factorize the probability of an associated DAG as

$$p_\theta(s|x) = \prod_{i \in I, j \in J} p_{\theta_i}(y_i \mid \text{Pa}(y_i)) \cdot p_{\theta_j}(z_j \mid \text{Pa}(z_j)), \tag{3}$$

where $\text{Pa}(\cdot)$ returns the parent nodes of a given node in the graph, and $\theta = \{\theta_k : k \in I \cup J\}$ denotes the parameter set of generative models in the compound AI system. This decomposition breaks down the likelihood of the system into a product of multiple terms, where each term contains

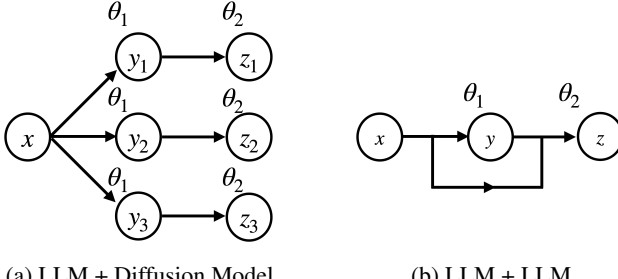

(a) LLM + Diffusion Model        (b) LLM + LLM

Figure 2: Corresponding DAGs of compound AI systems. **(a)** The user gives a prompt $x$ which is processed by the LLM $\theta_1$ to produce three captions $y_1, y_2, y_3$. The diffusion model $\theta_2$ is queried to generate images $z_i$ given $y_i$ for $i = 1, \ldots, 3$. **(b)** The user gives a prompt $x$ which is processed by the first LLM $\theta_1$ to produce an intermediate result $y$. Then, $x$ and $y$ are passed to the second model $\theta_2$ to generate the final output $z$.

a single model. Note that the labels of the models $\theta_i, \theta_j$ can refer to different queries to the same model. Taking the case of Figure 2 (a) as an example, and denoting the set of generated contents by $s = \{y_1, y_2, y_3, z_1, z_2, z_3\}$, we have $p(s|x) = \prod_{i=1}^{3} p_{\theta_1}(y_i|x) \cdot p_{\theta_2}(z_i|y_i)$.

**Preference Dataset Construction.** To learn the model parameters $\theta = \{\theta_k : k \in I \cup J\}$, we first construct a system-specific preference dataset. We assume that all the variables in (3) are observed. The dataset can be constructed in the following way: given an input $x$, the system generates two variants of the final outputs as well as intermediate outputs in the system. We label the preferred sample as $s^w = \{y_i^w \mid i \in I\} \cup \{z_j^w \mid j \in J\}$, and the non-preferred sample as $s^l = \{y_i^l \mid i \in I\} \cup \{z_j^l \mid j \in J\}$. Consider the case of Figure 2 (a) as an example, a preferred sample is in the form of $s^w = \{y_1^w, y_2^w, y_3^w, z_1^w, z_2^w, z_3^w\}$. Putting everything together, each preference data pair is composed of $(x, s^w, s^l)$. This construction process is tightly coupled with the structure of the underlying system. Different compound system architectures may involve different sets of intermediate variables, and thus require generating distinct preference datasets

**Loss Function Design.** Given the dataset $D$ composed of preference pairs $(x, s^w, s^l)$ and a compound AI system formulated as a DAG, we can apply the decomposition of (3) to the DPO loss (2):

$$L_{\text{Direct}}(\theta) = -\mathbb{E}_{(x, s^w, s^l) \sim \mathcal{D}} \left[ \log \sigma \left( \beta \log \frac{p_\theta(s^w|x)}{p_{\bar{\theta}}(s^w|x)} - \beta \log \frac{p_\theta(s^l|x)}{p_{\bar{\theta}}(s^l|x)} \right) \right], \quad (4)$$

where $\bar{\theta}$ denotes the collection of reference models, $\sigma(\cdot)$ stands for the sigmoid function. We can then find the optimal $\theta$ by simply optimizing the loss function (4), resulting in an end-to-end optimization.

### 2.3 SysDPO-Sampling

SysDPO-Direct requires system-specific datasets with observations of the intermediate outputs, whereas most existing preference datasets are composed of only inputs and ranked outputs. One redress is to reversely sample the intermediate outputs given input and final output to construct a semi-synthetic dataset. However, such a practice is costly and the samples might be of low quality. To this end, we introduce a variation of SysDPO, termed SysDPO-Sampling. The key difference between SysDPO-Sampling and SysDPO-Direct lies in how the probability $p_\theta(z|x)$ is decomposed.

The key idea of SysDPO-Sampling is as follows. Recall that $s := \{y_i, z_j\}_{i \in I, j \in J}$ is the set of all the variables generated by the system. Denote the collection of intermediate samples as $y := \{y_i\}_{i \in I}$, where $y \in \mathcal{Y}$. Thus, $s = \{z_j\}_{j \in J} \cup y$. Assuming a discrete sample space (e.g., discrete tokens), by the law of total probability:

$$p_\theta(\{z_j\}_{j \in J} \mid x) = \sum_{y \in \mathcal{Y}} p_\theta(s \mid x) = \sum_{y \in \mathcal{Y}} \prod_{i \in I, j \in J} p_{\theta_i}(y_i \mid \text{Pa}(y_i)) \cdot p_{\theta_j}(z_j \mid \text{Pa}(z_j)). \quad (5)$$

However, the summation of the right-hand side of (5) is generally intractable, as it requires summing over all possible sentences $y$. While one method is to use Monte Carlo sampling to approximate the expectation, it is inefficient if the sample space is large. To efficiently approximate the summation, we focus only on a small number of highly probable, distinct samples $y_i^\alpha$, given that the less probable samples contribute little to the summation. Therefore, having the highly probable distinct samples

$\{y_i^\alpha\}_{i,\alpha}$, we make the following approximation.

$$p_\theta(\{z_j\}_J|x) \approx \sum_\alpha \prod_{i \in I, j \in J} p_{\theta_i}(y_i^\alpha | \mathrm{Pa}(y_i^\alpha)) \cdot p_{\theta_j}(z_j | \mathrm{Pa}(z_j)). \tag{6}$$

To generate the samples indexed by $\alpha$, we employ Diverse Beam Search (DBS): an extension of standard beam search that enforces diversity across the beam groups by adding a penalty to candidates similar to those already selected within group partitions [34]. During training, these intermediate samples $\{y_i^\alpha\}_{i,\alpha}$ are regenerated after each model update step, while SysDPO-Direct is always trained on a fixed dataset. Plug this into the original DPO loss (2) and we arrive at the loss function $L_{\mathrm{Sampling}}$, as shown in Appendix B. Note that, in this case, end-to-end gradient-based optimization is feasible.

# 3 A Theoretical Analysis on Compound AI System Alignment

We ask a fundamental question: Does SysDPO lead to the correct alignment? To answer this question, we first define the concept of $\beta$-perfect alignment and show that, in the population setting, both regular DPO and RLHF lead to perfect alignment. Then, we prove that SysDPO similarly achieves $\beta$-perfect alignment on compound AI systems.

## 3.1 Perfect Alignment under Bradley-Terry Preference Model

Our theoretical analysis begins with defining the perfect alignment with respect to the preference oracle (1):

**Definition 1** ($\beta$-Perfect Alignment). *Define a probabilistic generative model $\theta^* : \mathcal{X} \to \mathcal{Z}$ associated with a generative distribution $p_{\theta^*}$. We say $\theta^*$ is $\beta$-perfectly aligned with parameter $\beta > 0$ to the preference oracle $\mathrm{pref}(\cdot)$ if $\forall z^w, z^l \in \mathcal{Z}, \ x \in \mathcal{X}$:*

$$\frac{\mathrm{pref}(z^w \succ z^l \mid x)}{\mathrm{pref}(z^l \succ z^w \mid x)} = \left(\frac{p_{\theta^*}(z^w \mid x)}{p_{\theta^*}(z^l \mid x)}\right)^\beta.$$

*An equivalent formulation is,*

$$\mathrm{pref}(z^w \succ z^l \mid x) = \frac{p_{\theta^*}(z^w \mid x)^\beta}{p_{\theta^*}(z^w \mid x)^\beta + p_{\theta^*}(z^l \mid x)^\beta}.$$

This definition has several interpretations. First, for all $\beta > 0$, the above definition satisfies the order consistency of the generative model $\theta^*$ and the preference oracle $\mathrm{pref}$, i.e., $\mathrm{pref}(z^w \succ z^l \mid x) > \mathrm{pref}(z^l \succ z^w \mid x)$ if and only if $p_{\theta^*}(z^w \mid x) > p_{\theta^*}(z^l \mid x)$. Then, with varying $\beta$, we can see that it acts as a temperature parameter controlling how peaked the aligned policy must be to match a given preference signal. Writing

$$\mathrm{pref}(z^w \succ z^l \mid x) = \left(1 + \left(\tfrac{p_{\theta^*}(z^l|x)}{p_{\theta^*}(z^w|x)}\right)^\beta\right)^{-1},$$

one sees that smaller $\beta$ forces larger likelihood ratios to represent the same preference score from the oracle (thus a more deterministic policy), while larger $\beta$ makes the required ratios closer to 1 (thus a near-uniform policy). When the policy is viewed as Boltzmann rational $p_{\theta^*}(z \mid x) \propto \exp(r^*(x, z)/\beta)$ with Bradley–Terry preferences $\mathrm{pref}(z^w \succ z^l \mid x) = \exp(r^*(x, z^w))/\big(\exp(r^*(x, z^w)) + \exp(r^*(x, z^l))\big)$ (e.g., see [12]), then Definition 1 holds exactly; in this view, $\beta$ is an inverse rationality parameter. Alternatively, if we view the generative model $p_{\theta^*}$ as a choice model, then it recovers Luce's choice axiom (independence from irrelevant alternatives) [17] when $\beta = 1$.

Moreover, the notion of perfect alignment has been implicitly used in prior work [26]. In the following, we show that given the preference oracle, we can find a $\beta$-perfectly aligned model by optimizing the DPO objective (2), or equivalently the following RLHF objective:

$$\max_{p_\theta} \quad \mathbb{E}_{x \sim \mathcal{D}_x, z \sim p_\theta(z|x)}[r^*(z, x)] - \beta \mathbb{D}_{\mathrm{KL}}(p_\theta \parallel p_{\bar\theta}), \tag{7}$$

where $\mathcal{D}_x$ is a distribution over $\mathcal{X}$; $\bar\theta$ is a reference model; and $\beta$ is the strength of KL regularization.

**Proposition 1.** *Suppose an optimal model $\theta^* \in \Theta$ achieves the maximum of the RLHF objective* (7) *or the minimum of the DPO loss function* (2) *where data preference distribution $\mathcal{D}$ is given by the preference oracle* pref$(\cdot)$. *Then, it follows that $\theta^*$ is $\beta$-perfectly aligned with the preference oracle* (1) *when the output of the reference model $\bar{\theta}$ follows a uniform distribution given any input $x \in \mathcal{X}$.*

The implication of the proposition is as follows. In ideal case where we have infinite ground truth preference data, there is complete information about perfect alignment. In such cases, the alignment objective should not rely on a reference model, which is why RLHF and DPO achieve perfect alignment when the reference model is simply a uniform distribution. However, the reference model has proven to be important in practice, where we only have insufficient preference data.

In the following, we investigate the proposed heuristics, SysDPO-Direct and SysDPO-Sampling, which also achieve perfect alignment.

### 3.2 SysDPO Achieves Perfect Alignment

SysDPO-Direct works by simply replacing the final output $z := \{z_j\}_{j \in J}$ in the original DPO loss function by the set of all generated variables $s := \{y_i\}_{i \in I} \cup z \in \mathcal{S}$. Then, the corresponding preference oracle for two sets is inherited from the original preference oracle as follows.

$$\text{pref}_{\text{sys}}(s^w \succ s^l \mid x) := \text{pref}(z^w \succ z^l \mid x).$$

In the following theorem, we show that the generative system $\theta^*_{\text{sys}}$ aligned by SysDPO-Direct is indeed $\beta$-perfect given the following technical assumption. Intuitively, the assumption demands diversity in the training distribution $\mathcal{D}$ used in SysDPO.

**Assumption 1.** *Any $s \in \mathcal{Y} \times \mathcal{Z}$ has a positive probability to be sampled from $\mathcal{D}$.*

**Theorem 1.** *Under Assumption 1, suppose an optimal model $\theta^*_{sys} \in \Theta$ achieves the minimum of the SysDPO loss function* (4) *where the preference for data is given by the preference oracle* pref$_{\text{sys}}$. *Then, $\theta^*_{sys}$ is $\beta$-perfectly aligned with the preference oracle* pref$(\cdot)$ *when the reference model follows a uniform distribution.*

Assumption 1 highlights a fundamental challenge of compound AI system alignment. In compound AI systems, intermediate outputs are typically hidden from the user, so preferences can only be given over the final outputs. Since the evaluation of a component in a compound system must depend on other components, there is no preference oracle to compare two intermediate outputs directly. Thus, SysDPO-Direct may benefit from a training dataset of diverse intermediate outputs.

In cases where the preference dataset lacks intermediate results, we propose an alternative variant of SysDPO, i.e., SysDPO-Sampling. It aims to directly optimize an approximated standard DPO loss function with respect to $p_\theta(z|x)$ through sampling. Therefore, at the population level where an infinite number of samples can be drawn, Proposition 1 implies optimality of SysDPO-Sampling. We further discuss the finite-sample setting through the lens of coresets in Appendix E, suggesting that SysDPO-Sampling may also benefit from a diverse sampling scheme.

## 4 Applications

In this section, we discuss two examples of compound AI systems corresponding to Figure 2. We demonstrate how SysDPO-Direct and SySDPO-Sampling can be applied to these examples.

### 4.1 An LLM and a Diffusion Model

We apply SysDPO-Direct to the example in Figure 1, which involves an LLM $\psi$ and a diffusion model $\phi$. Given an input prompt $x$ provided to the system, the LLM generates an intermediate output $y$, which can be parsed into multiple captions $y = (y_1, y_2, \ldots, y_n)$. Each $y_i$, $i = 1, \ldots, n$ serves as a prompt for the diffusion model. The diffusion model is then queried $n$ times, generating images $z_1, z_2, \ldots, z_n$ as the final outputs. Similarly to Figure 1, the generated images are expected to follow a logical relationship. This demands that both the language model and the diffusion model not only recognize their roles in the overall task but also execute them accurately and coordinate effectively to ensure coherent system behavior. As such, this setting serves as a strong testbed for evaluating our proposed method.

This multi-step process is modeled as a DAG whose special case ($n = 3$) is shown in Figure 2 (a), where we can decompose the generation process as $p(s|x) = p_\psi(y|x) \cdot \prod_{i=1}^{n} p_\phi(z_i|y_i)$. We start by applying the probability decomposition to the SysDPO-Direct loss function (4). However, although the LLM's generation likelihood $p_\psi(y|x)$ is accessible, the diffusion model's $p_\phi(z|y)$ is not. We address this challenge by extending [35] to accommodate our setting and obtain an upper bound of the SysDPO loss function. We then optimize this upper bound to align the system. Detailed derivation is elaborated in the Appendix F.

## 4.2 Compound LLM Collaboration System

We also explore systems formed purely by the collaboration of language models. In such systems, multiple LLMs cooperate to complete a complex task. Specifically, we study a two-stage question-answering system, where a user poses a question as input $x$, the first-stage model $\psi_1$ generates an intermediate answer $y$, and the second-stage model $\psi_2$ refines it to produce the final output $z$. This setup serves as a simple yet representative compound AI system to demonstrate how multiple LLMs can collaborate toward a shared objective. The overall generation process of the system can be formalized as:

$$p(s \mid x) = p_{\psi_1}(y \mid x) \cdot p_{\psi_2}(z \mid x, y),$$

where $s = \{y, z\}$ represents the intermediate and final outputs. Since both $\psi_1$ and $\psi_2$ are language models with tractable likelihoods, we can apply either SysDPO-Sampling or SysDPO-Direct to align the system. SysDPO-Direct requires a preference dataset that includes both the intermediate prompt (i.e., the output of $\psi_1$) and the final output, which can be constructed using a ground-truth reward model. In the absence of this constructed preference dataset, SysDPO-Sampling applies by only requiring a ready-made preference dataset that contains only input and final response pairs.

# 5 Experiments

## 5.1 Compound AI System of a LLM and a Diffusion Model

This section evaluates the effectiveness of SysDPO-Direct for aligning compound AI systems. Motivated by the example in Figure 1, we synthesize a preference dataset and experiment on the joint alignment of an LLM and a diffusion model. Examples of synthetic data are shown in the Appendix I. Our evaluation focuses on the coherence of the generated image sequences and their alignment with coherent system-level preferences.

**Dataset Construction.** We construct a custom dataset in three steps. First, we use the regressor from Zhuang et al. [43] to assign scores in $[0, 1]$ to images based on 40 scene-related attributes (e.g., brightness, coldness, and boring). These attributes cover a wide range of visual concepts and provide general, high-level labels that reflect diverse aspects of scene semantics. Second, for each attribute, GPT-4 is used to generate 250 user prompts instructing the system to produce image sequences with progressive changes. To increase prompt diversity, we adopt four prompt styles from Qin et al. [24]. Details are provided in Appendix H. Third, for each prompt, four sequences are generated and ranked using the Preference Score in Eq. (8). Six comparison pairs are constructed, with the higher scoring sequence in each pair labeled as preferred, resulting in 6,000 total comparisons.

**Preference Score.** To compare the generated image sequences, we define a *preference score $q$* that evaluates both the order consistency and the uniformity of the distribution. This metric is based on the attribute scores assigned to the images by the regressor from Zhang et al. [43]. Given a sequence of three images with attribute scores $a_1, a_2$, and $a_3$, the Preference Score $q$ is computed as:

$$q = -\big(a_1 - a_3 + |a_2 - (a_1 + a_3)/2|\big). \tag{8}$$

Sequences with higher $q$ values are preferred as they reflect correct ordering and smoother distributions. In contrast, reversed or uneven sequences result in lower $q$. Further details, including examples illustrating the calculation of $q$, are provided in Appendix G.

**Models.** For the construction and evaluation of the dataset, we use an instruction-tuned Llama-3-8B model [2]. To generate image sequences for constructing chosen and rejected samples in the dataset, we employ Stable Diffusion XL (SDXL) [23]. For training purposes, we use Stable Diffusion 1.5 [28] which provides a balance between computational efficiency and generation quality.

**Evaluation.** The performance of the system is evaluated using two metrics. The first metric is the **Average Preference Score** across all generated sequences from the test dataset. The second evaluation metric is the **Order Consistency Ratio**, measuring the proportion of generated sequences in the correct order, i.e., where $a_1 < a_2 < a_3$.

**Baselines.** To evaluate the effectiveness of the proposed SysDPO-Direct joint alignment approach, we compare it against four baseline methods. (1) System Before Alignment: system prior to applying SysDPO-Direct. Notably, Llama-3-8B is instruction-tuned, and it serves as a baseline for separately aligned systems. (2) Best-of-N Sampling: from four generated sequences per prompt, the one with the highest Preference Score is selected. (3) Only Train Language Model: the diffusion model is frozen, and only the language model is aligned using the dataset and the loss function of SysDPO-Direct. (4) Only Train Diffusion Model: the language model is frozen, and only the diffusion model is aligned. All baselines use the same dataset and loss.

**Results** We evaluate the system using the Preference Score and Order Consistency Ratio. Examples of system outputs before and after training can be found in the Appendix I.

The results in Table 1 demonstrate the importance of alignment in compound AI systems and the effectiveness of the proposed SysDPO-Direct alignment approach. The "System Before Alignment" baseline achieves poor performance, with a low Preference Score and a low Order Consistency Ratio (32%), indicating that conventionally instruction-tuned components are insufficient to ensure coherent collaboration required in compound systems. The "Only Train Language Model" baseline achieves significantly better results, with a Preference Score of 0.23 and a ratio of 65%. This highlights that the language model plays a critical role in guiding the overall behavior of the system, as it generates captions that directly influence the outputs of the diffusion model. In contrast, the performance gain from training only the diffusion model is inherently constrained by the captions produced by the fixed language model, thus its performance gain is notably lower than that of the Only Train Language Model baseline. SysDPO-Direct achieves the best Preference Score (0.25) and the highest Order Consistency Ratio (73%). These results validate the effectiveness of our SysDPO algorithm, demonstrating its ability to optimize both components together for superior performance in generating coherent and progressive image sequences. The training dynamics of all three methods are provided in Appendix J.

| Method | Pref. Score | OC Ratio |
|---|---|---|
| **SysDPO-Direct (Proposed)** | **0.25** | **73%** |
| System Before Alignment | -0.20 | 32% |
| Best-of-Sampling | 0.16 | 67% |
| Only Train Language Model | 0.23 | 65% |
| Only Train Diffusion Model | -0.03 | 38% |

Table 1: Performance comparison of the proposed method and baselines. Higher Preference Scores (Pref. Score) and higher Order Consistency Ratios (OC Ratio) are better.

| Method | WR-Chosen | WR-Prompted |
|---|---|---|
| **SysDPO-Sampling** | **19.8** | **66.4** |
| Prompted System | 12.8 | / |
| Separate-DPO | 16.6 | 57.3 |
| SysDPO-$\psi_1$ | 16.0 | 60.4 |
| SysDPO-$\psi_2$ | 18.1 | 63.9 |

Table 2: Overall Performance Comparison. WR-chosen denotes the win rate (%) against human-preferred responses in the dataset, and WR-prompted measures the win rate against the Prompted System baseline.

## 5.2 Compound LLM Collaboration System

This section aims to evaluate the performance of our joint alignment methods, SysDPO, in the two-stage LLM collaboration system described in Section 4.2. In our implementation, we employ two instances of `Qwen1.5-1.8B-Chat` [33] to serve as $\psi_1$ and $\psi_2$, respectively. The two models are trained without sharing parameters. This two-LLM configuration represents a typical and illustrative setting for analyzing system-level alignment dynamics, offering a clear view of inter-model coordination and preference optimization.

We further extend our framework to a three-LLM system, demonstrating its scalability to larger compound architectures. The setup and preliminary observations of this extension are presented in Appendix D. In the following sections, we focus on the two-LLM system as the main case study for detailed analysis and discussion.

**Dataset.** We employ the preference dataset `Intel/orca-dpo-pairs` [32] for DPO training, consisting of 129000 instructions paired with corresponding preference examples (each instruction has a pair of chosen/rejected responses). We sample 193 instruction data points as the evaluation set. The remaining examples are used for the training process. We directly use the dataset without generating intermediate outputs. Since no ground truth reward model is available to give preference scores to intermediate samples, we focus on evaluating SysDPO-Sampling.

**Evaluation.** We adopt the evaluator `weighted-alpaca-eval-gpt4-turbo` [15], an automatic annotator based on `gpt-4-turbo`, to assess model performance through pairwise comparisons. Given a pair of outputs—one from the evaluated system and one from a reference—the annotator assigns preference between the pair. The Win Rate is defined as the proportion of cases where the model output is judged superior to the reference output. To better understand the performance of the model from different perspectives, we report two types of Win Rate. The first, WR-chosen, uses the chosen response from the preference dataset as the reference, reflecting alignment with human-labeled preferences. The second, WR-Prompted uses a prompting-based baseline system as the reference, which will be introduced in detail later.

**Baselines.** We compare SysDPO-Sampling with several baseline approaches to evaluate its effectiveness in aligning compound AI systems. As a simple baseline, we explore the prompting-based composition of the two-stage system without any alignment or additional training. Specifically, we directly connect two models and give task-specific prompts to guide their collaboration. This setup is referred to as the Prompted System. The second baseline, Separate-DPO, follows a similar prompting scheme to coordinate the two models but introduces alignment by training each stage individually using the original DPO objective on the `Intel/orca-dpo-pairs` dataset. After alignment, the two stages are composed into a compound system. Each stage is optimized independently with its own preference signals, without joint training or system-level feedback. In contrast, SysDPO-Sampling jointly aligns the entire two-stage system using holistic preference signals. Rather than optimizing each component in isolation, it directly optimizes the composed system based on end-to-end preferences, allowing both stages to adapt cooperatively to user-aligned objectives.

**Training Details** For the results reported below, we sampled two intermediate outputs per step during training. An analysis of the effect of sample method is provided in Section 5.2.3. During both training and evaluation, we set the maximum token number at 256 in the sampling process. We trained the models using LoRA with $\beta = 0.5$, a learning rate of $1 \times 10^{-7}$, and an accumulated batch size of 128. Aligning the entire system took approximately 30 hours on a single NVIDIA H200 GPU.

### 5.2.1 Is joint alignment necessary?

Results in Table 2 show that the non-optimized compound AI system (Prompted System) performs the worst, underscoring the limitations of relying solely on prompt engineering for effective component coordination. In contrast, SysDPO-Sampling achieves substantial gains, improving the win rate against chosen responses from 12.8% to 19.8%—a 55% relative improvement. Its outputs are also preferred 66.4% of the time over those of the prompted system.

The Separate-DPO baseline represents a conventional approach in which each component is aligned individually, without considering preferences over the behavior of the composed system. Although this method yields better performance than unaligned prompting, it is still outperformed by our joint alignment method. Notably, in this setting, the collaboration between the two models is relatively simple, and their roles are similar. Yet, even under these favorable conditions for Separate-DPO, SysDPO-Sampling still achieves superior results. In more general scenarios, where the interaction between components is more complex, their roles are more distinct, or there are no clean training data available per component, the performance of Separate-DPO is likely to further degrade. These results underscore the value of optimizing compound systems holistically rather than in isolation.

### 5.2.2 How much does each stage contribute to the final performance?

To better understand how system-level alignment affects each component, we design two variants of SysDPO-Sampling where only one stage is updated during training, while the other is kept frozen. Importantly, the training is still guided by system-level preference signals, and the overall loss is computed using SysDPO-Sampling. In the SysDPO-$\psi_1$ setting, we train $\psi_1$ while freezing $\psi_2$; conversely, in SysDPO-$\psi_2$, we train $\psi_2$ and freeze $\psi_1$. As shown in Table 2, SysDPO-$\psi_1$

achieves 16.0% WR-Chosen and 60.4% WR-Prompted, while SysDPO-$\psi_2$ achieves 18.1% and 63.9%, respectively. These results indicate that both components benefit from system-level alignment, but $\psi_2$ plays a more critical role in determining the final quality of the output. This is likely because $\psi_2$ directly generates the final response and has access to both the input $x$ and the intermediate output $y$. Crucially, both variants outperform the Prompted System, indicating that during joint training, system-level preferences are effectively distributed across components, and each model is able to learn and adapt accordingly. However, neither of them matches the full SysDPO-Sampling system, emphasizing that jointly optimizing both components leads to better performance and more effective collaboration.

We observe that SysDPO-Sampling converges more slowly than its stage-wise variants but achieves the highest performance; detailed training dynamics are provided in Appendix K.

### 5.2.3 Effect of Finite Sample Approximation on Performance

In Section 2.3, we approximate the loss function by sampling only a small number of highly probable and distinct candidates $y_i^\alpha$. In this section, we discuss how this sample set is generated and analyze how the sampling strategy and sample size affect training performance.

Our main experiments are based on the Diverse Beam Search (DBS) method [34], which generates $k$ diverse candidates per input. We compare DBS with standard Monte Carlo (MC) sampling and summarize the results in Table 3.

During training, we sample $k$ intermediate candidates $y^1, \ldots, y^k \sim p_{\psi_1}(\cdot \mid x)$ and use them to compute the loss. At evaluation time, we set the temperature to $0$ and deterministically selecting the most probable outputs.

For DBS, we obtain $k$ candidates using $k$ beam groups and a diversity penalty of 20, reporting results for $k = 2, 4$. For MC sampling, we independently draw $k$ samples with temperature $1.0$ under different random seeds, evaluating $k \in \{2, 3, 4, 5\}$.

| Method | Sample Size | WR-Prompted (%) |
|--------|-------------|-----------------|
| DBS | 2 | 68.5 |
| DBS | 4 | 68.2 |
| MC | 2 | 66.8 |
| MC | 3 | 67.0 |
| MC | 4 | 66.7 |
| MC | 5 | 66.0 |

Table 3: Comparison between Diverse Beam Search (DBS) and Monte Carlo (MC) sampling under different sample sizes $k$.

Table 3 summarizes the win rate (WR-Prompted) for each configuration. DBS consistently outperforms MC with the same number of sampled candidates. Both methods show stable performance as $k$ varies, suggesting that as few as two samples may suffice without significantly increasing computational costs. Compared with DBS, MC often produces near-duplicate candidates, as illustrated by qualitative examples in Appendix L. The training dynamics are shown in Appendix M.

## 6  Conclusion and Discussion

We propose a principled framework for aligning compound AI systems by modeling them as DAGs and optimizing for system-level preferences. Our two methods, SysDPO-Direct and SysDPO-Sampling, address settings with and without intermediate output, showing both theoretical and empirical gains across two compound tasks. While demonstrated on specific systems, our approach has broader potential in domains like healthcare and education, where complex multi-component AI workflows require careful alignment to ensure safety and usability.

Despite these advances, several open challenges remain. For instance, extending SysDPO to systems with more components and complex interactions is a natural next step. Another key direction is improving training efficiency. SysDPO-Direct requires access to intermediate generations and may become expensive when components produce high-dimensional or latent outputs (e.g., in vision or multimodal systems). Exploring approximations or sample-efficient estimators would be beneficial. For SysDPO-Sampling, better sampling strategies could improve gradient estimation when intermediate states are unobserved. Moreover, the broader design space of compound AI systems, such as dynamic routing, feedback loops, or interactive collaboration, raises new alignment questions that extend beyond static DAGs. Future work should explore how to generalize SysDPO to settings with non-static structures and latent coordination mechanisms.

## Acknowledgments

SK acknowledges support by NSF 2046795 and 2205329, IES R305C240046, ARPA-H, the MacArthur Foundation, Schmidt Sciences, HAI, OpenAI, Microsoft, and Google.

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

# Appendix

## A  Complete Prompts in Figure 1

The complete prompts generated by GPT-4 for DALL-E to generate the six images in Figure 1 are as follows.

---

**Left picture**

(a) A realistic image of a calm cat sitting comfortably on a soft cushion. The cat has a serene expression, with its eyes half-closed in contentment. It has fluffy fur, predominantly grey with white patches, and its tail is neatly curled around its body. The background is a cozy room with a hint of sunlight filtering through a window, casting a warm glow on the cat.

(b) A realistic image of a slightly irritated cat sitting on a wooden floor. The cat has a slight frown and its ears are tilted back slightly, indicating mild annoyance. It has short fur, a mix of black and white, and is staring directly at the viewer with wide eyes. The background is simple, with a few scattered cat toys and a small plant, adding to the homey atmosphere.

(c) A realistic image of a very angry cat with its fur bristled and ears flattened against its head. The cat's eyes are narrowed and glowing, mouth slightly open in a hiss. It has thick, long fur that is mostly black with some orange stripes. The background shows a stormy sky through a window, enhancing the dramatic mood. The cat is on a dark rug, which contrasts with its bright, intense eyes.

---

**Right picture**

(a) A cartoon-style image of a cat with a slightly annoyed expression. The cat has fluffy white fur, bright green eyes, and its ears are slightly pulled back. It is sitting comfortably on a soft blue cushion, with its tail flicking lightly to the side. The background is a simple, soft yellow to keep the focus on the cat's expression.

(b) A cartoon-style image of a cat with an angry expression. The cat has fluffy grey fur, narrowed yellow eyes, and its ears are flat against its head. It is standing with an arched back on a wooden floor, with its fur bristling and its claws slightly out. The background is a dimly lit room, adding to the menacing atmosphere around the cat.

(c) A cartoon-style image of a cat with a furious expression. The cat has short black fur, glaring red eyes, and its ears are pinned back. It is hissing aggressively, showing sharp teeth, with a puffed tail. The cat stands on a stormy night background, with lightning in the sky, enhancing the dramatic and intense mood.

---

# B  Details of the Loss Function of SysDPO-Sampling $L_{sampling}$

To approximate the system-level preference probabilities in Equation (2), we apply the sampling-based approximation in Equation (6), which rewrites the system likelihood as a product over component-level probabilities.

Plugging this into the original DPO loss, we obtain the following training objective:

$$
L(\theta) = -\mathbb{E}_{(x,z^w,z^l)\sim\mathcal{D}}\left[ \log \sigma \left( \beta \log \frac{\sum_\alpha \left[ \prod_{i\in I} p_{\theta_i}(y_i^\alpha | \operatorname{Pa}(y_i^\alpha)) \prod_{j\in J} p_{\theta_j}(z_j^w | \operatorname{Pa}(z_j^w)) \right]}{\sum_\alpha \left[ \prod_{i\in I} p_{\bar\theta_i}(y_i^\alpha | \operatorname{Pa}(y_i^\alpha)) \prod_{j\in J} p_{\bar\theta_j}(z_j^w | \operatorname{Pa}(z_j^w)) \right]} \right.\right.
$$

$$
\left.\left. - \beta \log \frac{\sum_\alpha \left[ \prod_{i\in I} p_{\theta_i}(y_i^\alpha | \operatorname{Pa}(y_i^\alpha)) \prod_{j\in J} p_{\theta_j}(z_j^l | \operatorname{Pa}(z_j^l)) \right]}{\sum_\alpha \left[ \prod_{i\in I} p_{\bar\theta_i}(y_i^\alpha | \operatorname{Pa}(y_i^\alpha)) \prod_{j\in J} p_{\bar\theta_j}(z_j^l | \operatorname{Pa}(z_j^l)) \right]} \right) \right] \tag{9}
$$

This formulation allows us to perform end-to-end gradient-based optimization over all model components $\{\theta_i\}$ and $\{\theta_j\}$ in a compound AI system. To generate the samples indexed by $\alpha$, we employ Diverse Beam Search [34], which still selects the highest-probability sample after applying the diversity penalty. This ensures that the sampled trajectories are both diverse and high-probability, which aligns well with our approximation.

# C  Theoretical Analysis

*Proof of Proposition 1.* Given the reward function $r^*$, it follows from prior work [26, 22, 21, 14] that the solution $\pi_{\theta^*}$ to (7) satisfies

$$
r^*(x,z) = \beta \log \frac{p_{\theta^*}(z\mid x)}{p_{\bar\theta}(z\mid x)} + \beta \log G(x), \tag{10}
$$

with $G(x) = \int_z p_{\bar\theta}(z\mid x)\exp(\beta^{-1}r^*(z,x))\mathrm{d}z$ being the partition function. Then, we plug (10) into (1), with some algebraic manipulation, we arrive at

$$
\operatorname{pref}(z^w \succ z^l \mid x) = \frac{\left(\frac{p_{\theta^*}(z^w|x)}{p_{\bar\theta}(z^w|x)}\right)^\beta}{\left(\frac{p_{\theta^*}(z^w|x)}{p_{\bar\theta}(z^w|x)}\right)^\beta + \left(\frac{p_{\theta^*}(z^l|x)}{p_{\bar\theta}(z^l|x)}\right)^\beta}.
$$

When the reference model $\bar p$ is a uniform distribution, we conclude

$$
\operatorname{pref}(z^w \succ z^l \mid x) = \frac{p_{\theta^*}(z^w\mid x)^\beta}{p_{\theta^*}(z^w\mid x)^\beta + p_{\theta^*}(z^l\mid x)^\beta},
$$

showing that $\theta^*$ is perfectly aligned (Definition 1).

Next, let us examine what is a solution to the DPO loss function (2). Given that the reference model is a uniform distribution, the DPO loss is simplified to

$$
\begin{aligned}
L(\theta) &= -\mathbb{E}_{(x,z^w,z^l)\sim D}\left[ \log \sigma \left( \beta \log p_\theta(z^w|x) - \beta \log p_\theta(z^l|x) \right) \right] \\
&= -\mathbb{E}_{(x,z^w,z^l)\sim D}\left[ \log \sigma \left( \beta \log \frac{p_\theta(z^w|x)}{p_\theta(z^l|x)} \right) \right] \\
&= -\mathbb{E}_{(x,z^w,z^l)\sim D} \log \left[ \frac{p_\theta(z^w|x)^\beta}{p_\theta(z^w|x)^\beta + p_\theta(z^l|x)^\beta} \right]. \tag{11}
\end{aligned}
$$

Let us review the definition of this preference data distribution $\mathcal{D}$. Given any data triplet $(x,z,z') \in \mathcal{X} \times \mathcal{Z} \times \mathcal{Z}$ sampled from a pre-preference data generation process $\mathcal{D}'$, the preference oracle would

label $(z^w = z, z^l = z')$ with probability $\text{pref}(z \succ z' \mid x)$, and label $(z^w = z', z^l = z)$ with probability $1 - \text{pref}(z \succ z' \mid x)$. Therefore, denoting

$$\text{pref}_\theta(z \succ z' \mid x) := \frac{p_\theta(z|x)^\beta}{p_\theta(z|x)^\beta + p_\theta(z'|x)^\beta},$$

the DPO loss above can be written as

$$
\begin{aligned}
(11) = -\mathbb{E}_{(x,z,z') \sim D'} [ & \text{pref}(z \succ z' \mid x) \log \text{pref}_\theta(z \succ z' \mid x) \\
& + (1 - \text{pref}(z \succ z' \mid x)) \log \text{pref}_\theta(z' \succ z \mid x)] \\
= -\mathbb{E}_{(x,z,z') \sim D'} [ & \text{pref}(z \succ z' \mid x) \log \text{pref}_\theta(z \succ z' \mid x) \\
& + (1 - \text{pref}(z \succ z' \mid x)) \log (1 - \text{pref}_\theta(z \succ z' \mid x))].
\end{aligned}
$$

Noticing that the above is precisely a cross entropy term, we conclude that the minimum is achieved when $\theta^*$ satisfies

$$\text{pref}(z \succ z' \mid x) = \text{pref}_{\theta^*}(z \succ z' \mid x) = \frac{p_{\theta^*}(z|x)^\beta}{p_{\theta^*}(z|x)^\beta + p_{\theta^*}(z'|x)^\beta}.$$

Thus, the solution $\theta^*$ to the DPO loss function agrees with $\beta$-perfect alignment.

$\square$

*Proof of Theorem 1.* SysDPO-Direct works by simply replacing the final output $z$ in the original DPO loss function by the set of all generated variables $s := \{y_i\}_{i \in I} \cup \{z\}$. By definition, the corresponding preference oracle for two sets is inherited from the original preference oracle as follows.

$$\text{pref}_{\text{sys}}(s^w \succ s^l \mid x) := \text{pref}(z^w \succ z^l \mid x), \tag{12}$$

where $z^w \in s^w$ and $z^l \in s^l$.

Thus, Proposition 1 implies that if $\theta^*_{\text{sys}}$ minimizes the SysDPO loss function (4) with uniform prior, it satisfies

$$\frac{\text{pref}_{\text{sys}}(s^w \succ s^l \mid x)}{\text{pref}_{\text{sys}}(s^l \succ s^w \mid x)} = \left( \frac{p_{\theta^*_{\text{sys}}}(s^w \mid x)}{p_{\theta^*_{\text{sys}}}(s^l \mid x)} \right)^\beta.$$

Applying (12) and rearranging the above equality, we have

$$
\begin{aligned}
\left( \frac{\text{pref}(z^w \succ z^l \mid x)}{\text{pref}(z^l \succ z^w \mid x)} \right)^{1/\beta} &= \left( \frac{\text{pref}_{\text{sys}}(s^w \succ s^l \mid x)}{\text{pref}_{\text{sys}}(s^l \succ s^w \mid x)} \right)^{1/\beta} = \frac{p_{\theta^*_{\text{sys}}}(s^w \mid x)}{p_{\theta^*_{\text{sys}}}(s^l \mid x)} \\
&= \frac{p_{\theta^*_{\text{sys}}}(\{y_i^w\}_{i \in I} \cup \{z^w\} \mid x)}{p_{\theta^*_{\text{sys}}}(\{y_i^l\}_{i \in I} \cup \{z^l\} \mid x)}.
\end{aligned}
\tag{13}
$$

A key observation from the above equations is: the LHS is independent from any $y_i$. Thus, there should be some freedom in what $y_i$ we include on the RHS. Concretely, Assumption 1 states that, for any intermediate samples $\forall \{y_i\}_{i \in I} \in \mathcal{Y}$ and final output $z \in \mathcal{Z}$, we know $(\{y_i\}_{i \in I}, z)$ appear with positive probability in the SysDPO loss. Recall minimizing the SysDPO loss require (13) holds for all pairs $(s^w, s^l)$ sampled from $\mathcal{D}$. Thus, the above equation must hold for $\forall \{y_i\}_{i \in I} \in \mathcal{Y}$ : $(\{\{y_i\}_{i \in I}, z^w\}, \{\{y_i\}_{i \in I}, z^l\})$. Thus, it implies that the RHS is also independent from any $y_i$.

This result can be intuitively interpreted in the following way. Consider the assumptions that the training data $\mathcal{D}$ is good enough and the optimal model $\theta^*_{\text{sys}}$ is capable enough to achieve the minimum of the SysDPO loss. Then, the optimal model learns to extract sufficient information from all those possible intermediate outputs $y_i$ such that it achieves perfect alignment no matter what $y_i$ is sampled from the data distribution.

Given this observation, we can simply replace $\{y_i^w\}_{i \in I}$ and $\{y_i^l\}_{i \in I}$ in (13) by any $\{y_i\}_{i \in I} \in \mathcal{Y}$ without changing the value of (13). Rearranging the equality and sum $\{y_i\}_{i \in I}$ over its sample space

$\mathcal{Y}$ on both sides:

$$(13) \cdot p_{\theta_{\text{sys}}^*}(\{y_i\}_{i \in I} \cup \{z^l\} \mid x) = p_{\theta_{\text{sys}}^*}(\{y_i\}_{i \in I} \cup \{z^w\} \mid x)$$

$$\implies (13) \cdot \sum_{\mathcal{Y}} p_{\theta_{\text{sys}}^*}(\{y_i\}_{i \in I} \cup \{z^l\} \mid x) = \sum_{\mathcal{Y}} p_{\theta_{\text{sys}}^*}(\{y_i\}_{i \in I} \cup \{z^w\} \mid x)$$

$$\implies (13) \cdot p_{\theta_{\text{sys}}^*}(z^w \mid x) = p_{\theta_{\text{sys}}^*}(z^l \mid x)$$

$$\implies (13) = \frac{p_{\theta_{\text{sys}}^*}(z^w \mid x)}{p_{\theta_{\text{sys}}^*}(z^l \mid x)}.$$

Raising the above terms to the power of $\beta$, we recover the definition of perfect alignment. $\square$

## D  Extend to a three-LLM system

To further examine the scalability of our framework, we extend the two-LLM configuration described in Section 5.2 to a three-LLM system. In this setup, first-stage LLMs $\psi_1$ and $\psi_2$ independently generate responses to the same input query, and the second stage LLM $\psi_3$ serves as a synthesizer that aggregates and refines these intermediate outputs into a final answer. This architecture introduces an additional component of coordination and preference alignment across multiple generation components, allowing us to test the framework's ability to align compound systems beyond pairwise interactions.

We train this system using the same SysDPO procedure employed in the main experiments, without modifying any hyper-parameters. Evaluation is performed with the WR-Prompted metric, directly comparing the post-alignment system with its pre-alignment counterpart. The aligned three-LLM system achieves a 58 % win rate against the unaligned version, suggesting that the proposed method effectively generalizes to larger multi-component architectures.

While large-scale experiments involving many components are currently constrained by our computational budget, these preliminary results provide encouraging evidence that our framework offers a solid foundation for future scaling toward more complex compound AI systems.

## E  Discussion on Finite-sample analysis for SysDPO-Sampling

In this section, we discuss how a diverse sampling scheme may benefit SysDPO-Sampling through a two-generative-model collaboration system (Figure 2 (b)).

As shown in Section 2.3, SysDPO-Sampling aims to approximate the intractable generation probability $p_\theta(z \mid x) = \sum_{y \in \mathcal{Y}} p_{\theta_2}(z \mid y, x) p_{\theta_1}(y \mid x)$ with finite samples $\widehat{\mathcal{Y}} := \{y^\alpha\}_\alpha$ such that approximately $p_\theta(z \mid x) \propto \sum_\alpha p_{\theta_2}(z \mid y^\alpha, x) p_{\theta_1}(y^\alpha \mid x)$.

This can be viewed as a coreset selection problem. I.e., the goal is to find a diverse subset $\widehat{\mathcal{Y}}$ of representative points to approximate the full set $\mathcal{Y}$. Denoting $\mu_{\mathcal{Y}} := \frac{1}{|\mathcal{Y}|} \sum_{y \in \mathcal{Y}} p_{\theta_2}(\cdot \mid y, x) p_{\theta_1}(y \mid x)$ (similarly for $\mu_{\widehat{\mathcal{Y}}}$), we aim to minimize the approximation error $\epsilon := |\mu_{\mathcal{Y}} - \mu_{\widehat{\mathcal{Y}}}|$.

Therefore, a strategically selected subset often outperform random Monte-Carlo (MC) sampled subset, where the strategy typically involves selecting representative and diverse points [5, 41, 29]. Diverse Beam Search (DBS) can be seen as a greedy coreset-selection algorithm where, at every decoding step, it keeps the subset that jointly maximizes a combined objective of generation likelihood and diversity.

The core of a finite-sample analysis is the rate of approximation error $\epsilon$ w.r.t. the budget $k := |\widehat{\mathcal{Y}}|$. Although it is unclear what the exact error $\epsilon$ of DBS is, coreset algorithms generally achieve a worst-case guarantee of $\epsilon = \mathcal{O}(\frac{1}{\sqrt{k}})$ [5]. For example, consider finite sample space $\mathcal{Z}$, and denote the sampled probability mass function as $v_y := p_{\theta_2}(\cdot \mid y, x) p_{\theta_1}(y \mid x) \in \mathbb{R}^{|\mathcal{Z}|}$. Assume all $v_y$ is $\ell_2$-bounded by $D$. The approximate carathéodory theory [18] states that we can always find a subset $\widehat{\mathcal{Y}} \subset \mathcal{Y}$ of size $k$ such that the approximation error $\epsilon = \mathcal{O}(\frac{D}{\sqrt{k}})$. To make the bound tighter, we need fine-grained analysis that exploits the concrete structure of the distribution, which is an exciting but

non-trivial future work. Nonetheless, in practice, coreset can be significantly better as shown in prior works [5, 41, 29] as well as our experiments.

# F Deriving the Loss function of LLM + Diffusion System

In this section of the appendix, we provide a detailed explanation of the DDPM diffusion model, and the derivation of the loss function used for this system.

## F.1 Denoising Diffusion Probabilistic Model (DDPM)

DDPM [10] is a widely used class of diffusion model. Below is a highlight of the key ingredient we need for DPO for DDPM [35], with our framework.

Given a real image $z_0$, consider a diffusion process, which we call the forward process, gradually making the original image into Gaussian noise $z_T$ after $T$ steps, i.e.,

$$z_0 \to z_1 \to z_2 \to \cdots \to z_T \sim \mathcal{N}(0, I).$$

The goal of the diffusion model $\phi$ is to reverse this process that recovers an image from noise. The forward process and the reverse process are denoted respectively as

$$q(z_{0:T}|y), \qquad p_\phi(z_{0:T}|y),$$

where $y$ is the context, i.e., the prompt to the diffusion model.

Note that both the forward and backward processes are Markovian, and in particular we have the nice property that the forward process

$$q(z_{0:T}|y) = q(z_0|y) \prod_{t=1}^{T} q(z_t|z_{t-1}), \qquad \text{where each } q(z_t|z_{t-1}) \text{ is a Gaussian.}$$

Similarly, the reverse process

$$p_\phi(z_{0:T}|y) = p(z_T) \prod_{t=1}^{T} p_\phi(z_{t-1}|z_t, y), \qquad \text{where each } p_\phi(z_{t-1}|z_t, y) \text{ is a Gaussian.} \tag{14}$$

In this formulation, the ideal goal for the diffusion model is that $q(z_{0:T}|y) = p_\phi(z_{0:T}|y)$. However, this is not easy to optimize directly. With some analysis, the DDPM paper [10] proposes to minimize for

$$D_{KL}(q(z_{t-1}|z_t, z_0, y)\|p_\phi(z_{t-1}|z_t, y)) \quad \text{for} \quad t \sim \mathcal{U}([T]), z_0 \sim q(z_0|y),$$

where $\mathcal{U}(\cdot)$ denotes the uniform distribution on a set, and $[T]$ denotes the set of $\{1, 2, \ldots, T\}$. This is done by learning a denoiser $\epsilon_\phi$ operating in the following way. For a real image $z_0 \sim q(z_0|y)$, we sample noise $\epsilon \sim \mathcal{N}(0, I)$, and have

$$z_t(z_0, \epsilon) = \sqrt{\bar{\alpha}_t} z_0 + \sqrt{1 - \bar{\alpha}_t} \epsilon, \tag{15}$$

where $\bar{\alpha}_t$ is some parameter such that $z_t \sim q(z_t|z_0)$. Then, the denoiser predicts the noise $\epsilon$ that is added to the $z_0$. I.e.,

$$\epsilon_\phi(z_t(z_0, \epsilon), t, y) \text{ aims to predict } \epsilon.$$

The denoiser $\epsilon_\phi$ is essentially a reparameterization of the mean of $p_\phi(z_{t-1}|z_t, y)$.

The **key ingredient** is that, as shown in [10],

$$D_{KL}(q(z_{t-1}|z_t, z_0, y)\|p_\phi(z_{t-1}|z_t, y)) = \mathbb{E}_{\epsilon \sim \mathcal{N}(0,I)} \left[ w_t \left\| \epsilon - \epsilon_\phi(z_t(z_0, \epsilon), t, y) \right\|^2 \right] + C, \tag{16}$$

where $w_t$ is a weight parameter and $C$ is a constant independent of model $\phi$.

Therefore, modeling $\epsilon_\phi$ by a neural net, the DDPM model $\phi$ is trained to minimize the above objective averaged over samples of $y, z_0, \epsilon, t$.

## F.2 Dealing with the Diffusion Model in SysDPO

Recall in the main text we obtain the System DPO loss function as:

$$L(\psi, \phi) = -\mathbb{E}_{(x,s^w,s^l)\sim D}\left[\log\sigma\left(\beta\left(\log\frac{p_\psi(y^w|x)}{p_{\bar\psi}(y^w|x)} + \sum_i^n \log\frac{p_\phi(z_i^w|y_i^w)}{p_{\bar\phi}(z_i^w|y_i^w)}\right)\right.\right.$$
$$\left.\left. - \beta\left(\log\frac{p_\psi(y^l|x)}{p_{\bar\psi}(y^l|x)} + \sum_i^n \log\frac{p_\phi(z_i^l|y_i^l)}{p_{\bar\phi}(z_i^l|y_i^l)}\right)\right)\right].$$

The next step is to convert the likelihood of the diffusion model $p_\phi$ to something optimizable.

Consider the generated image as the whole process, i.e.,

$$z_{i,0:T} := \{z_{i,0}, z_{i,1}, \ldots, z_{i,T}\},$$

where $z_{i,0}$ is the generated image, while the others are things in the middle. Following the same notation, we denote

$$z_{i,t-1,t} := \{z_{i,t-1}, z_{i,t}\}.$$

The preference is considered to be given to the every processes that generates $z_0$ as the end outcome. Following [35], we have

$$L(\theta, \phi) = -\mathbb{E}_{(x,s^w,s^l)\sim D}\left[\log\sigma\left(\beta\mathbb{E}_{z_{i,1:T}^w\sim q(z_{i,1:T}^w|z_{w,0}^i), z_{i,1:T}^l\sim q(z_{i,1:T}^l|z_{l,0}^i)}\right.\right.$$
$$\left.\left.\left(\left(\log\frac{p_\theta(y_w|x)}{p_{\bar\theta}(y_w|x)} + \sum_i \log\frac{p_\phi(z_{w,0:T}^i|y_w^i)}{p_{\bar\phi}(z_{w,0:T}^i|y_w^i)}\right) - \left(\log\frac{p_\theta(y_l|x)}{p_{\bar\theta}(y_l|x)} + \sum_i \log\frac{p_\phi(z_{l,0:T}^i|y_l^i)}{p_{\bar\phi}(z_{l,0:T}^i|y_l^i)}\right)\right)\right)\right].$$

Recall the decomposition of the reverse process (14), we have

$$L(\theta, \phi) = -\mathbb{E}_{(x,s^w,s^l)\sim D}\left[\log\sigma\left(\beta\mathbb{E}_{z_{i,1:T}^w\sim q(z_{i,1:T}^w|z_{w,0}^i), z_{i,1:T}^l\sim q(z_{i,1:T}^l|z_{l,0}^i)}\right.\right.$$
$$\left.\left.\left(\left(\log\frac{p_\theta(y_w|x)}{p_{\bar\theta}(y_w|x)} + \sum_i\sum_{t=1}^T \log\frac{p_\phi(z_{w,t-1}^i|z_{i,t}^w, y_w^i)}{p_{\bar\phi}(z_{w,t-1}^i|z_{i,t}^w, y_w^i)}\right) - \left(\log\frac{p_\theta(y_l|x)}{p_{\bar\theta}(y_l|x)} + \sum_i\sum_{t=1}^T \log\frac{p_\phi(z_{l,t-1}^i|z_{i,t}^l, y_l^i)}{p_{\bar\phi}(z_{l,t-1}^i|z_{i,t}^l, y_l^i)}\right)\right)\right)\right].$$

Note that $\sum_{t=1}^T = T\mathbb{E}_{t\sim\mathcal{U}([T])}$ for $t$ being a random variable uniformly distributed on $1, 2, \ldots, T$. Simply denoting $\mathbb{E}_{t\sim\mathcal{U}([T])}$ as $E_t$, we have

$$L(\theta, \phi) = -\mathbb{E}_{(x,s^w,s^l)\sim D}\left[\log\sigma\left(\beta\mathbb{E}_{z_{i,1:T}^w\sim q(z_{i,1:T}^w|z_{w,0}^i), z_{i,1:T}^l\sim q(z_{i,1:T}^l|z_{l,0}^i)}\right.\right.$$
$$\left.\left.\left(\left(\log\frac{p_\theta(y_w|x)}{p_{\bar\theta}(y_w|x)} + T\sum_i \mathbb{E}_t\log\frac{p_\phi(z_{w,t-1}^i|z_{i,t}^w, y_w^i)}{p_{\bar\phi}(z_{w,t-1}^i|z_{i,t}^w, y_w^i)}\right) - \left(\log\frac{p_\theta(y_l|x)}{p_{\bar\theta}(y_l|x)} + T\sum_i \mathbb{E}_t\log\frac{p_\phi(z_{l,t-1}^i|z_{i,t}^l, y_l^i)}{p_{\bar\phi}(z_{l,t-1}^i|z_{i,t}^l, y_l^i)}\right)\right)\right)\right]$$
$$= -\mathbb{E}_{(x,s^w,s^l)\sim D}\left[\log\sigma\left(\beta\mathbb{E}_{z_{i,1:T}^w\sim q(z_{i,1:T}^w|z_{w,0}^i), z_{i,1:T}^l\sim q(z_{i,1:T}^l|z_{l,0}^i)}\mathbb{E}_t\right.\right.$$
$$\left.\left.\left(\left(\log\frac{p_\theta(y_w|x)}{p_{\bar\theta}(y_w|x)} + T\sum_i \log\frac{p_\phi(z_{w,t-1}^i|z_{i,t}^w, y_w^i)}{p_{\bar\phi}(z_{w,t-1}^i|z_{i,t}^w, y_w^i)}\right) - \left(\log\frac{p_\theta(y_l|x)}{p_{\bar\theta}(y_l|x)} + T\sum_i \log\frac{p_\phi(z_{l,t-1}^i|z_{i,t}^l, y_l^i)}{p_{\bar\phi}(z_{l,t-1}^i|z_{i,t}^l, y_l^i)}\right)\right)\right)\right].$$

Next, we may further simplify the equation by switching $\mathbb{E}_{z_{i,1:T}^w \sim q(z_{i,1:T}^w | z_{w,0}^i), z_{i,1:T}^l \sim q(z_{i,1:T}^l | z_{l,0}^i)}$ and $\mathbb{E}_t$ in the above, i.e.,

$$
L(\theta, \phi) = -\mathbb{E}_{(x,s^w,s^l) \sim D} \Bigg[ \log \sigma \Bigg( \beta \mathbb{E}_t \, \mathbb{E}_{z_{i,1:T}^w \sim q(z_{i,1:T}^w | z_{w,0}^i), z_{i,1:T}^l \sim q(z_{i,1:T}^l | z_{l,0}^i)}
$$

$$
\left( \left( \log \frac{p_\theta(y_w|x)}{p_{\bar\theta}(y_w|x)} + T \sum_i \log \frac{p_\phi(z_{w,t-1}^i | z_{i,t}^w, y_w^i)}{p_{\bar\phi}(z_{w,t-1}^i | z_{i,t}^w, y_w^i)} \right) - \left( \log \frac{p_\theta(y_l|x)}{p_{\bar\theta}(y_l|x)} + T \sum_i \log \frac{p_\phi(z_{l,t-1}^i | z_{i,t}^l, y_l^i)}{p_{\bar\phi}(z_{l,t-1}^i | z_{i,t}^l, y_l^i)} \right) \right) \Bigg) \Bigg]
$$

$$
= -\mathbb{E}_{(x,s^w,s^l) \sim D} \Bigg[ \log \sigma \Bigg( \beta \mathbb{E}_t \, \mathbb{E}_{z_{w,t-1,t}^i \sim q(z_{w,t-1,t}^i | z_{w,0}^i), z_{l,t-1,t}^i \sim q(z_{l,t-1,t}^i | z_{l,0}^i)}
$$

$$
\left( \left( \log \frac{p_\theta(y_w|x)}{p_{\bar\theta}(y_w|x)} + T \sum_i \log \frac{p_\phi(z_{w,t-1}^i | z_{i,t}^w, y_w^i)}{p_{\bar\phi}(z_{w,t-1}^i | z_{i,t}^w, y_w^i)} \right) - \left( \log \frac{p_\theta(y_l|x)}{p_{\bar\theta}(y_l|x)} + T \sum_i \log \frac{p_\phi(z_{l,t-1}^i | z_{i,t}^l, y_l^i)}{p_{\bar\phi}(z_{l,t-1}^i | z_{i,t}^l, y_l^i)} \right) \right) \Bigg) \Bigg]
$$

The rationale for the above can be illustrated as follows. Consider a random variables $Z_1, \ldots, Z_T$, and any function $f : (Z_{t-1}, Z_t) \to \mathbb{R}$ for any $t \in [T]$. Then, denoting $\delta_s^t$ as the indicator function, i.e., $\delta_s^t = 1$ only if $s = t$, and $\delta_s^t = 0$ otherwise, we can derive

$$
\mathbb{E}_{Z_{1:T}} \, \mathbb{E}_{t \sim \mathcal{U}([T])} \, f(Z_{t-1}, Z_t) = \mathbb{E}_{Z_{1:T}} \, \mathbb{E}_{t \sim \mathcal{U}([T])} \sum_{s=1}^T \delta_s^t \cdot f(Z_{s-1}, Z_s)
$$

$$
= \mathbb{E}_{t \sim \mathcal{U}([T])} \sum_{s=1}^T \delta_s^t \cdot \mathbb{E}_{Z_{1:T}} f(Z_{s-1}, Z_s)
$$

$$
= \mathbb{E}_{t \sim \mathcal{U}([T])} \sum_{s=1}^T \delta_s^t \cdot \mathbb{E}_{Z_{s-1}, Z_s} f(Z_{s-1}, Z_s)
$$

$$
= \mathbb{E}_{t \sim \mathcal{U}([T])} \cdot \mathbb{E}_{Z_{t-1}, Z_t} f(Z_{t-1}, Z_t).
$$

Next, noting that $q(z_{w,t-1,t}^i | z_{w,0}^i) = q(z_{i,t}^w | z_{w,0}^i) \cdot q(z_{w,t-1}^i | z_{w,0}^i, z_{i,t}^w)$ (similarly for $q(z_{l,t-1,t}^i | z_{l,0}^i)$), we can first sample $z_{i,t}^w$ and then $z_{w,t-1}^i$ separately, i.e.,

$$
L(\theta, \phi) = -\mathbb{E}_{(x,s^w,s^l) \sim D} \Bigg[ \log \sigma \Bigg( \beta \mathbb{E}_t \, \mathbb{E}_{z_{i,t}^w \sim q(z_{i,t}^w | z_{w,0}^i), z_{i,t}^l \sim q(z_{i,t}^l | z_{l,0}^i)} \, \mathbb{E}_{z_{w,t-1}^i \sim q(z_{w,t-1}^i | z_{w,0}^i, z_{i,t}^w), z_{l,t-1}^i \sim q(z_{l,t-1}^i | z_{l,0}^i, z_{i,t}^l)}
$$

$$
\left( \left( \log \frac{p_\theta(y_w|x)}{p_{\bar\theta}(y_w|x)} + T \sum_i \log \frac{p_\phi(z_{w,t-1}^i | z_{i,t}^w, y_w^i)}{p_{\bar\phi}(z_{w,t-1}^i | z_{i,t}^w, y_w^i)} \right) - \left( \log \frac{p_\theta(y_l|x)}{p_{\bar\theta}(y_l|x)} + T \sum_i \log \frac{p_\phi(z_{l,t-1}^i | z_{i,t}^l, y_l^i)}{p_{\bar\phi}(z_{l,t-1}^i | z_{i,t}^l, y_l^i)} \right) \right) \Bigg) \Bigg].
$$

Since $-\log \sigma$ is convex, by Jensen's inequality, we have

$$
L(\theta, \phi) \leq -\mathbb{E}_{(x,s^w,s^l) \sim D} \mathbb{E}_t \, \mathbb{E}_{z_{i,t}^w \sim q(z_{i,t}^w | z_{w,0}^i), z_{i,t}^l \sim q(z_{i,t}^l | z_{l,0}^i)} \Bigg[ \log \sigma \Bigg( \beta \mathbb{E}_{z_{w,t-1}^i \sim q(z_{w,t-1}^i | z_{w,0}^i, z_{i,t}^w), z_{l,t-1}^i \sim q(z_{l,t-1}^i | z_{l,0}^i, z_{i,t}^l)}
$$

$$
\left( \left( \log \frac{p_\theta(y_w|x)}{p_{\bar\theta}(y_w|x)} + T \sum_i \log \frac{p_\phi(z_{w,t-1}^i | z_{i,t}^w, y_w^i)}{p_{\bar\phi}(z_{w,t-1}^i | z_{i,t}^w, y_w^i)} \right) - \left( \log \frac{p_\theta(y_l|x)}{p_{\bar\theta}(y_l|x)} + T \sum_i \log \frac{p_\phi(z_{l,t-1}^i | z_{i,t}^l, y_l^i)}{p_{\bar\phi}(z_{l,t-1}^i | z_{i,t}^l, y_l^i)} \right) \right) \Bigg) \Bigg].
$$
$$
\tag{17}
$$

Recall that we we have been done so far are all for making the diffusion model's log probability efficiently computable. To complete the derivation, it left to convert the log-probabilities to the

denoising loss via (16). Specifically, with $C$ being the constant appears in (16), we can see that

$$\mathbb{E}_{z^i_{w,t-1}\sim q(z^i_{w,t-1}|z^w_{w,0},z^w_{i,t})}\log\frac{p_\phi(z^i_{w,t-1}|z^w_{i,t},y^i_w)}{p_{\bar\phi}(z^i_{w,t-1}|z^w_{i,t},y^i_w)}$$

$$=\mathbb{E}_{z^i_{w,t-1}\sim q(z^i_{w,t-1}|z^w_{w,0},z^w_{i,t})}\left(\log\frac{p_\phi(z^i_{w,t-1}|z^w_{i,t},y^i_w)}{q(z^i_{w,t-1}|z^i_{w,0},z^w_{i,t})}-\log\frac{p_{\bar\phi}(z^i_{w,t-1}|z^w_{i,t},y^i_w)}{q(z^i_{w,t-1}|z^i_{w,0},z^w_{i,t})}\right)$$

$$=-D_{KL}(q(z^i_{w,t-1}|z^i_{w,0},z^w_{i,t})\|p_\phi(z^i_{w,t-1}|z^w_{i,t},y^i_w))+D_{KL}(q(z^i_{w,t-1}|z^i_{w,0},z^w_{i,t})\|p_{\bar\phi}(z^i_{w,t-1}|z^w_{i,t},y^i_w))$$

$$=-D_{KL}(q(z^i_{w,t-1}|z^i_{w,0},z^w_{i,t})\|p_\phi(z^i_{w,t-1}|z^w_{i,t},y^i_w))+C+D_{KL}(q(z^i_{w,t-1}|z^i_{w,0},z^w_{i,t})\|p_{\bar\phi}(z^i_{w,t-1}|z^w_{i,t},y^i_w))-C$$

$$=-\mathbb{E}_{\epsilon\sim\mathcal{N}(0,I)}\left[w_t\left\|\epsilon-\epsilon_\phi(z_t(z^i_{w,0},\epsilon),t,y^i_w)\right\|^2\right]+\mathbb{E}_{\epsilon\sim\mathcal{N}(0,I)}\left[w_t\left\|\epsilon-\epsilon_{\bar\phi}(z_t(z^i_{w,0},\epsilon),t,y^i_w)\right\|^2\right].$$

To simplify the notation, we denote

$$\ell_\epsilon(\phi;t,z^w_{i,t},y^i_w):=\left[w_t\left\|\epsilon-\epsilon_\phi(z^w_{i,t},t,y^i_w)\right\|^2\right],$$

where the $\epsilon$ corresponds to the noise from which $z^w_{i,t}$ is derived (see (15)). Similarly, we use $\ell_\epsilon(\phi;t,z^l_{i,t},y^i_l)$ to denote the denoising loss for the losing data.

Thus we can write (17) as

$$L(\theta,\phi)\le-\mathbb{E}_{(x,s^w,s^l)\sim D}\mathbb{E}_t\,\mathbb{E}_{z^w_{i,t}\sim q(z^w_{i,t}|z^i_{w,0}),z^l_{i,t}\sim q(z^l_{i,t}|z^i_{l,0})}$$

$$\left[\left[\log\sigma\left(\beta\left(\left(\log\frac{p_\theta(y_w|x)}{p_{\bar\theta}(y_w|x)}+T\sum_i(-\ell_\epsilon(\phi;t,z^w_{i,t},y^i_w)+\ell_\epsilon(\bar\phi;t,z^w_{i,t},y^i_w))\right)\right.\right.\right.$$

$$\left.\left.\left.-\left(\log\frac{p_\theta(y_l|x)}{p_{\bar\theta}(y_l|x)}+T\sum_i(-\ell_\epsilon(\phi;t,z^l_{i,t},y^i_l)+\ell_\epsilon(\bar\phi;t,z^l_{i,t},y^i_l))\right)\right)\right)\mid t\right]\right].$$

Thus, we obtain a tractable loss function for SysDPO-Direct.

Since only approximate log-probabilities of the diffusion model are available, SysDPO-Sampling is not applicable in this setting.

## G   Preference Score Calculation

**Definition.**   The Preference Score $q$ evaluates the quality of a sequence of three images with attribute scores $a_1, a_2$, and $a_3\in[0,1]$, and is computed as:

$$q=-\left(a_1-a_3+\left|a_2-\frac{a_1+a_3}{2}\right|\right)$$

**Properties.**   The Preference Score reflects two aspects: **(1) Order Consistency:** A correctly ordered sequence ($a_1<a_2<a_3$) yields a higher $q$ value, while a reversed sequence results in a lower $q$ value. **(2) Distribution Evenness:** A sequence where $a_2$ is closer to the midpoint between $a_1$ and $a_3$ maximizes the score.

**Example Calculation.**   Consider four sequences of attribute scores:

- Sequence $\mathbf{a}=[1,0.5,0]$
- Sequence $\mathbf{b}=[0,1,0.9]$
- Sequence $\mathbf{c}=[0,0.5,1]$

For $\mathbf{a}$:

$$q_a=-\left(1-0+\left|0.5-\frac{1+0}{2}\right|\right)=-1$$

For **b**:

$$q_b = -\left(0 - 0.9 + \left|1 - \frac{0 + 0.9}{2}\right|\right) = 0.35$$

For **c**:

$$q_b = -\left(0 - 1 + \left|0.5 - \frac{0 + 1}{2}\right|\right) = 1$$

Since $q_a < q_b$, sequence **b** is preferred between sequence **a** and **b**. Sequence **c** is preferred between sequence **b** and **c**. This illustrates how the Preference Score penalizes uneven intermediate distributions or incorrect orderings.

## H  Prompt Styles and Examples

To ensure diversity in user prompts, we utilize four distinct prompt styles inspired by [24]. Each style varies in how it frames the objective for image generation. For illustration, all the examples below are based on the attribute "bright," showcasing how this attribute can be expressed in different styles.

### H.1  Prompt Styles

**Prompt-Based Style.**  This style of prompt directly describes the objective to be generated. It provides a clear and concise target for the system. For example:

> A series of images showing a garden with increasing brightness, from dawn to midday.

**Instruction-Based Style.**  This style uses instructional language to explicitly direct the system on what to generate. The phrasing is structured as a command or directive. For example:

> Generate a series of images of a morning scene, increasing the brightness and cheerfulness.

**Inspiration-Based Style.**  This style reflects a user's desire or inspiration for what they want to see. The prompt is expressed as a personal request or imaginative wish. For example:

> I want to see a series of images of a mountain as it gets progressively brighter.

**Hypothesis-Based Style.**  This style frames the generation task as a hypothetical scenario, often using conditional or reasoning-based language. The prompt includes both the condition and the desired outcome. For example:

> If the scene becomes brighter, the series of images will show progressively more illuminated scenes.

## I  Examples of Input and Output of Experiments

This appendix presents illustrative examples of the system's inputs and outputs. Each example consists of a user prompt as input and a sequence of three images as the output. Examples include:

(1) Success cases before training, highlighting potential alignment issues in untrained systems.

(2) Failed cases before training, showcasing issues with order consistency or uneven attribute progression.

(3) Improved cases after training, demonstrating the effectiveness of the proposed SysDPO alignment method.

## I.1 Success Case Before Training

**Input Prompt**

I want to see a series of images of a beach, with varying intensity of sunlight from overcast to bright sun.

**Attribute**

Sunny

**Intermediate Captions**

(a) Generate an image of a serene beach on a cloudy day, with the sun hidden behind thick grey clouds, casting a soft, diffused light on the sand and sea. The waves are calm, and a few seagulls are flying overhead.

(b) Create an image of the same beach, but with the sun breaking through the clouds, casting a warm, golden light on the sand and sea. The waves are slightly choppy, and a few beach umbrellas and sun loungers are scattered along the shore.

(c) Produce an image of the beach on a bright, sunny day, with the sun high in the sky and shining brightly down on the sand and sea. The waves are crashing against the shore, and a group of people are enjoying the day, with some playing beach volleyball and others swimming in the ocean.

**Final Output Images**

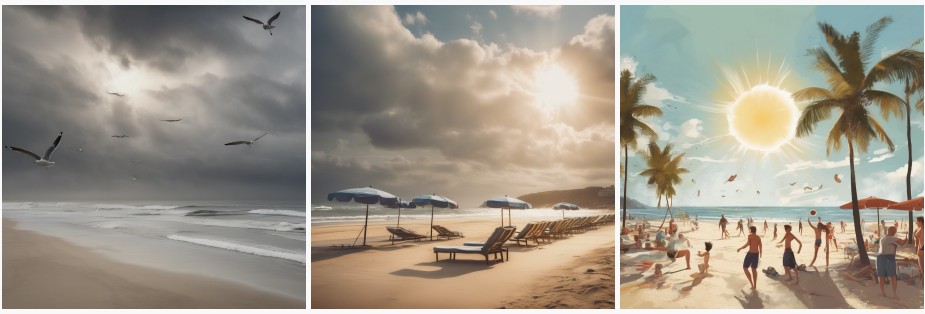

Figure 3: An example of a success output before training.

## I.2 Failed Case Before Training

**Input Prompt**

I want to see a series of images of a lake as the ice increases.

**Attribute**

Ice

**Intermediate Captions**

(a) Generate an image of a serene lake on a warm summer day, with calm water and lush greenery surrounding it, but with no signs of ice or frost.

(b) Create an image of the same lake, but with a thin layer of ice starting to form on the surface, with ripples and imperfections, and the surrounding landscape showing subtle signs of autumnal change, such as fallen leaves and fading greenery.

(c) Produce an image of the lake completely frozen, with a thick layer of ice covering its surface, and the surrounding landscape transformed into a winter wonderland, with snow-covered trees, frozen waterfalls, and a crisp, blue-gray sky.

**Final Output Images**

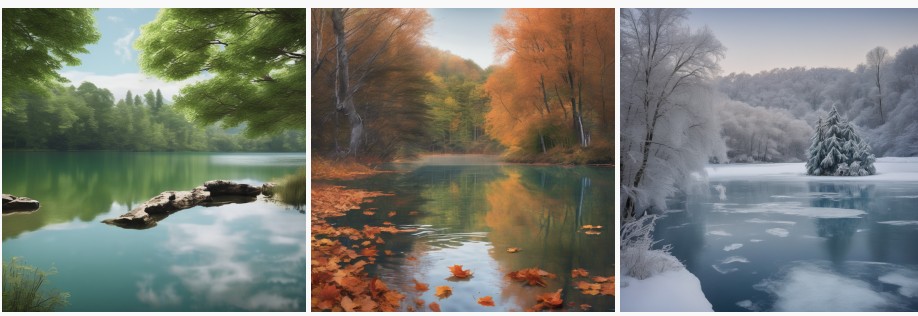

Figure 4: An example of a failure case before training. The first and second images lack clear progression in ice intensity.

### I.3 Improved Case After Training

**Input Prompt**

Create a series of images of a lake, increasing the thickness of ice.

**Attribute**

Ice

**Intermediate Captions**

(a) Generate a serene summer day with a calm and clear lake shoreline, featuring lush green trees and blooming vegetation, with water as the main focus. The water should be smooth and reflective, with a few boats or sailboats in the background. Ignore the water conditions.

(b) Continue with the same shoreline and surroundings, but with the onset of autumn, add gentle winds and decreasing temperatures. The vegetation should be dormant, and while the water is still calm, start to notice a slight decrease in clarity. There should be no ice yet, but hint at the changing conditions with minimal foliage or bare trees along the shore.

(c) In the dead of winter, depict a frozen landscape with the same shoreline, but now with thick, rugged ice covering the entire lake. The trees should be bare and shown as silhouettes against the harsh winter sky. The goal is a highly realistic, textured ice surface with no signs of water, and incorporate any relevant winter elements like snow-covered ground or bare branches.

**Final Output Images**

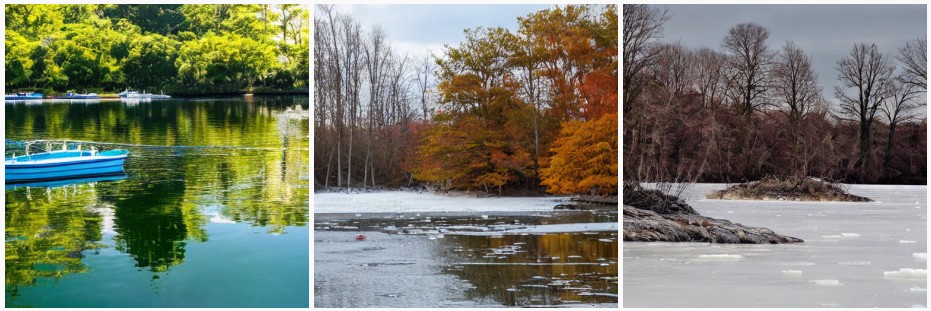

Figure 5: An example of an improved case after training. The sequence shows smooth and consistent progression in the ice intensity.

## J Training Dynamics of the Compound AI System with an LLM and a Diffusion Model

We present the training dynamics of SysDPO-Direct to illustrate how the joint optimization of the language model and the diffusion model improves system performance over time. Figure 6 shows the evolution of the Order Consistency Ratio throughout training, which exhibits steady and consistent improvements.

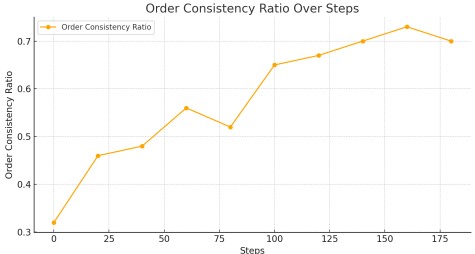

Figure 6: Order Consistency Ratio of SysDPO-Direct over training steps in Application 1. The consistent upward trend demonstrates the effectiveness of joint optimization in improving system-level coherence.

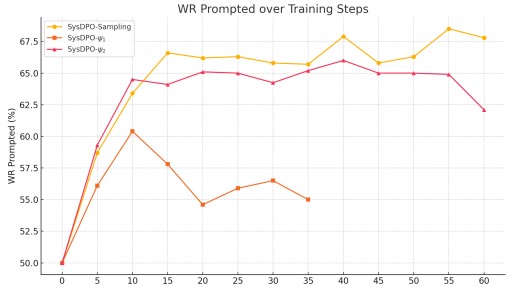

Figure 7: Training Dynamics of Stage-wise and Joint Alignment. WR-Prompted scores over training steps for SysDPO-Sampling, SysDPO-$\psi_1$, and SysDPO-$\psi_2$. Stage-wise alignment strategies converge more quickly, but SysDPO-Sampling ultimately achieves the highest performance.

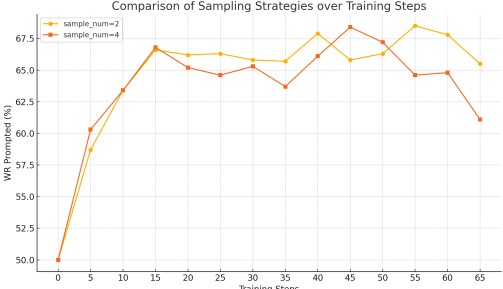

Figure 8: Comparison of Sampling Strategies During Training. WR-Prompted scores over training steps using two sampling strategies: diverse beam sampling with two candidates, and log-likelihood-guided contrastive selection from four candidates. Both strategies achieve comparable final performance, suggesting that sampling two diverse candidates is sufficient.

## K Training Dynamics of the Compound LLM Collaboration System

To better understand the learning behavior of different alignment strategies, we present the training dynamics of the two-stage LLM collaboration system in Figure 7. The curves show the WR-Prompted scores (%) evaluated periodically during training for SysDPO-Sampling and its two stage-wise variants, SysDPO-$\psi_1$ and SysDPO-$\psi_2$.

We observe that SysDPO-$\psi_1$ reaches its peak performance first, around step 10, but quickly exhibits noticeable fluctuations thereafter. This suggests that while $\psi_1$ can rapidly adapt to system-level supervision, its progress is constrained by the fixed, untrained $\psi_2$, limiting overall stability and further improvement.

SysDPO-$\psi_2$ shows a more gradual increase in performance, reaching its peak around step 40 and maintaining a more stable trajectory. This may be attributed to the role of $\psi_2$ as the final decision maker, which benefits from direct access to both the input $x$ and the intermediate response $y$, and thus can learn more steadily from the system-level feedback.

In contrast, SysDPO-Sampling improves more slowly during the early training steps, reflecting the higher complexity of jointly optimizing two interdependent models. However, it continues to improve throughout training and eventually achieves the highest performance, reaching its peak at around step 55. This demonstrates that joint training, although more challenging, enables both components to co-adapt and better coordinate under shared supervision, leading to superior final results.

After reaching their respective peaks, all three methods experience a slight decline in performance. A possible explanation for this trend is provided by recent findings [25, 9, 27], which show that running DPO for too long can lead to overoptimization, causing a decline in evaluation performance. They further observe that smaller models tend to degrade more quickly, whereas larger models show more stable behavior. Since our system is built upon 1.8B-parameter models, some degradation is observed in later stages of training.

## L  Diverse Examples of Intermediate Samples

To complement Table 3, we provide qualitative examples of intermediate responses produced by different sampling strategies for the same inputs. Consistent with our quantitative results, Diverse Beam Search (DBS) yields semantically distinct candidates that support more informative preference learning, whereas Multinomial (MC) sampling often produces near-duplicates that dilute the learning signal.

**Case 1: Extractive QA with abstention option.**

---

**Prompt**

Extract the answer to the following question from the movie plot. If the question isn't answerable, please output "Can't answer".
Question: What does Johanna cut from Katniss's arm?
Title: The Hunger Games: Catching Fire Movie
plot: After winning the 74th Hunger Games, Katniss Everdeen (Jennifer Lawrence) and Peeta Mellark (Josh Hutcherson) return home to District 12. President Snow visits Katniss at her home. The two make an agreement to not lie to one another, and Snow explains that her actions in the Games have inspired rebellions across the districts. He orders her to use the upcoming victory tour to convince him that her actions were out of genuine love for Peeta, not defiance against the Capitol, otherwise Katniss's loved ones will be killed. He shows her the clip where Gale kisses her as a warning that they are watching her. As the tour begins, Haymitch Abernathy, Katniss and Peeta's mentor, warns them that the "show" of their relationship must continue for the rest of their lives. Katniss suggests a public engagement between herself and Peeta, which is carried out and approved by Snow at his mansion in the Capitol. Returning home, Katniss warns her friend Gale Hawthorne of Snow's threat to kill both their families. Peacekeepers crack down on District 12, and Gale is publicly whipped after attacking new Head Peacekeeper Romulus Thread. Snow announces that the upcoming 75th Hunger Games, the Third Quarter Quell, will feature tributes selected from previous victors. Katniss immediately devotes herself to ensuring Peeta's survival, and recruits Haymitch's help to do so. At the reaping, Effie Trinket draws Katniss and Haymitch's names, but Peeta immediately volunteers to take Haymitch's place. As they prepare, Haymitch reveals that all the tributes are angry about being returned to the Games and advises a reluctant Katniss to make allies. In the pre-Games interviews, Katniss wears a wedding dress, as ordered by President Snow, but her stylist Cinna rigs it to transform into a representation of a mockingjay. Peeta announces that he and Katniss have secretly married and are expecting a baby, which causes outrage, and the audience of Capitol citizens begins shouting for the Games to be stopped, to no avail. Just before Katniss enters the...
Answer:

---

**Intermediate candidates produced by DBS sampling**

$y^1$: Haymitch Abernathy's arm.

$y^2$: Can't answer. The question is not answerable as it does not provide a specific scene or moment in the movie where Johanna cuts from Katniss's arm. The plot of the movie focuses on the events leading up to the 75 . . .

---

**Intermediate candidates produced by MC sampling**

$y^1$: Haymitch Abernathy.

$y^2$: Haymitch Abernathy.

**Case 2: Data-to-text generation.**

**Prompt**

Here is some data about a restaurant: name = The Golden Palace, eatType = coffee shop, food = French, priceRange = £20–25, customer rating = high, area = city centre. Write a sentence that includes the following data about a restaurant.

**Intermediate candidates produced by DBS sampling**

$y^1$: The Golden Palace is a fancy coffee shop in the city centre that serves delicious French food for a price range of £20–25, and it has a high customer rating, making it a great place to enjoy a meal with friends or family.

$y^2$: There is a restaurant called The Golden Palace located right in the city centre, where they serve delicious French cuisine at a price range of £18–22 per person, and customers have given it a high rating for its excellent service and cozy atmosphere.

**Intermediate candidates produced by MC sampling**

$y^1$: The Golden Palace is a fancy coffee shop located in the heart of the city center, serving satisfied customers who love dining there.

$y^2$: The Golden Palace is a fancy coffee shop located in the heart of the city centre, where they serve delicious French cuisine at a moderate price range of £20–25 per person, and customers highly rate their experience.

## M Effect of Sampling process in the Compound LLM Collaboration System

To examine how the sampling process influences training behavior and final performance, we compare two strategies for generating intermediate samples used in the loss function 9 in the two-stage LLM collaboration setting.

In the first strategy, we sample two intermediate candidate per input using diverse beam search [34]. We set the diversity penalty to 20 in our experiments. In the second strategy, we sample four intermediate responses for each input and then select a pair based on contrastive preferences. Specifically, we compute the log-likelihood of the final output for each intermediate response using the second-stage model, and choose the pair for which one sample leads to the highest likelihood of being preferred and the other the lowest. This contrastive selection aims to encourage sharper preference signals during training by focusing updates on clearly distinguishable pairs.

We further compare the training dynamics of the two sampling strategies by plotting WR-Prompted scores over training steps. As shown in Figure 8, both methods exhibit similar performance throughout training. This suggests that sampling only two candidates when combined with a sufficiently high diversity penalty is sufficient to yield informative preference pairs. Additionally, we observe that the contrastive selection strategy exhibits slightly greater fluctuations during training, potentially due to the higher variance introduced by selecting extreme pairs based on model log-likelihoods.

