# OpenReview forum: "Aligning Compound AI Systems via System-level DPO"
_NeurIPS.cc/2025/Conference — NeurIPS 2025 poster_

### Official Review · Reviewer_yR8i · 2025-07-02

**Clarity:** 3
**Significance:** 2
**Originality:** 2
**Rating:** 4
**Confidence:** 3

**Summary:**

This paper proposes a way to align compound AI systems, where multiple generative models are used to provide an output for a given input. The authors provide two simple extensions of direct preference optimization (DPO), a standard alignment technique, for compound AI systems. The first approach, SysDPO-Direct, assumes that we have a preference dataset with full intermediate outputs of the system. Then, instead of treating the preference as the comparison between two final outputs, SysDPO-Direct treats the preference as given to the comparison between two sets of intermediate and final outputs. Then, the rest of the training process is identical to DPO. The second approach, SysDPO-Sampling, is a method for cases where we do not have intermediate outputs in the preference dataset. It relies on a sample approximation of the probability of the output conditioned on the input by using high-probability samples. The validity of SysDPO-Direct—the same optimal policy as the standard DPO—has been proved. Two empirical studies show the usefulness of the proposed approach.

**Questions:**

Please refer to the weakness section. In particular,

- How are the samples in SysDPO-Sampling selected?
- How does their quality influence the final performance?
- How much better does the proposed approach perform against the trivial extension of DPO with a single Monte Carlo sample?

**Ethical Concerns:**

["NO or VERY MINOR ethics concerns only"]

**Final Justification:**

During the rebuttal phase, the authors provided a comparison with very natural baselines that I suggested in the first review, and successfully showed promising results over the baseline. The proposed approach will be a good baseline for future research in this direction. The reason for the current score (not a higher score) is that the experiments are limited in showing the goodness of the proposed approach as a general framework.

**Limitations:**

yes

**Paper Formatting Concerns:**

No.

**Quality:**

2

**Strengths And Weaknesses:**

Strength:

- The problem tackled in this paper is an recently important topic.
- The proposed approach is simple and natural. It is easy to understand.
- The validity of SysDPO-Direct has been theoretically shown, as well as the requirement (Assumption 1), which may be nontrivial.

Weakness:

- Because the approach is very simple, its technical innovation is limited. The approach of SysDPO-Direct is probably one of the approaches that everyone tries.
- It was not mentioned clearly how the intermediate samples are generated for SysDPO-Samples. It was also not clear how their quality affects the final performance.
- Only two experiments are conducted to show the usefulness of the proposed approach. Moreover, a trivial baseline is missing: DPO with a single sample Monte-Carlo to approximate (5).
- Related works are not mentioned. Are there any papers addressing the alignment of compounded systems?

---

> ### Author Rebuttal · Authors · 2025-07-31
>
> We sincerely thank the reviewer for the constructive and encouraging feedback. We will incorporate clarifications and revisions, as well as new experiments, in the camera-ready version. Details regarding your comments follow.
>
> - **Novelty and related works.**
>     - We thank the reviewer for the positive assessment and the request to clarify novelty and related work. Our contribution is **the first framework that *jointly aligns the parameters* of an entire compound AI system, modelled as a Directed Acyclic Graph (DAG).** This formulation lets us derive a DPO-style loss, and we prove the first alignment guarantee for compound AI systems. Different from our scope of parameter-level alignment, prior work primarily explores prompt-based coordination, which is an orthogonal line of research. It is an exciting future work to investigate how the two orthogonal directions interplay with each other, and how to effectively combine them.
>     - The resulting algorithm being conceptually simple is a result of its generality: it can be applied to any compound AI systems with a DAG factorization. Although the general scheme is simple, our implementations involve non-trivial procedures such as a tailored derivation of the tractable loss function for the LLM+Diffusion model system (Appendix D), and diverse sampling strategies used in both SysDPO-direct (Appendix F) and SysDPO-Sampling (Appendix B). Without such diversity-encouraged sampling procedures, MC sampling yields sub-optimal performance, as shown next.
> - **How the intermediate samples are generated and how their quality affects the final performance.** We would like to thank the reviewer for suggesting a more detailed explanation, as this is indeed a key factor for SysDPO-Sampling. We use Diverse Beam Search (DBS) for SysDPO Sampling (detailed in Appendix B), with a goal to improve diversity among the intermediate samples. We conduct a new ablation study to illustrate that the quality and diversity of those samples matter a lot. We begin by comparing Diverse Beam Search (DBS) with Monte‑Carlo (MC) sampling and then examine how performance scales with the number of intermediate candidates sampled during training. Our experiments reveal two clear trends: (i) within the tested range, increasing the sampling budget $k$ has little to no impact on overall performance; and (ii) DBS achieves higher win‑rates under the same sample count.
>     - **Experimental Settings and sample details**. Building on the LLM collaboration system in section 5.2, each training instance starts from input $x$, model $\psi_1$, and $\psi_2$. During training, we sample $k$ intermediate candidates $y^1, \dots y^k \sim p_{\psi_1}(\cdot | x)$ and leverage them to calculate the loss that jointly optimize $\psi_1, \psi_2$. At evaluation time, we eliminate sampling stochasticity by setting the temperature to 0 and deterministically selecting $y^\* = \max_y p_{\psi_1}(y | x)$  and $z^\* = \max_z p_{\psi_2}(z | x)$. We evaluate the system performance based on the quality of $z^\*$.
>     - **Sampling methods.** For DBS, We obtain $k$ candidates $y^1, \dots, y^k$ using $k$ beam groups and a diversity penalty of 20; we report results for $k=2,4$. For MC sampling, we independently sample $y^1, \dots, y^k$ with temperature 1.0 under different random seeds, evaluating $k=2,3,4,5.$
>     - **Results.** The table below summarizes the win‑rate of each configuration. **DBS outperforms MC with fewer number of sampled candidates.** Both the methods are consistent with varying $k$ within the test range, **suggesting that 2 samples may be sufficient without drastically increasing the budget.** Compared to DBS, MC often draws candidates that are near‑duplicates.
>
>
>         | Method | k | WR-Prompted (%) |
>         | --- | --- | --- |
>         | DBS | 2 | 68.5 |
>         | DBS | 4 | 68.2 |
>         | MC | 2 | 66.8 |
>         | MC | 3 | 67.0 |
>         | MC | 4 | 66.7 |
>         | MC | 5 | 66.0 |
>     - **Examples of the intermediate samples.** Here are some examples of intermediate outputs generated using different sampling methods for the same questions.
>         - Case 1:
>
>         > *Question:*  “Extract the answer to the following question from the movie plot. If the question isn't answerable, please output "Can't answer ……”:
>
>         > *DBS Sampling:*
>         >
>         >  $y^1$: Haymitch Abernathy's arm.
>         >
>         >  $y^2$: Can't answer. The question is not answerable as it does not provide a specific scene or moment in the movie where Johanna cuts from Katniss's arm. The plot …
>
>         >
>         > *MC  Sampling:*
>         >
>         > $y^1$: Haymitch Abernathy.
>         >
>         > $y^2$: Haymitch Abernathy.
>         >
>         - Case 2:
>
>         > *Question:* Here is some data about a restaurant: name = The Golden Palace, eatType = coffee shop, food = French, priceRange = £20-25, customer rating = high, area = city centre. Write a sentence that includes the following data about a restaurant
>
>         > *DBS Sampling:*
>         >
>         > $y^1$: The Golden Palace is a fancy coffee shop in the city centre that serves delicious French food for a price range of £20-25, and it has a high customer rating, making it a great place to enjoy a meal with friends or family.
>         >
>         > $y^2$: There is a restaurant called The Golden Palace located right in the city centre, where they serve delicious French cuisine at a price range of £18-22 per person, and customers have given it a high rating for its excellent service and cozy atmosphere.
>
>         > *MC Sampling:*
>         >
>         > $y^1$: The Golden Palace is a fancy coffee shop located in the heart of the city center, serving delicious French cuisine for around £20-25 per person, with a high rating from satisfied customers who love dining there.
>         >
>         > $y^2$: The Golden Palace is a fancy coffee shop located in the heart of the city centre, where they serve delicious French cuisine at a moderate price range of £20-25 per person, and customers highly rate their experience.
>         >
>
>         The MC samples are nearly identical, whereas DBS supplies two semantically distinct options. This illustrates how DBS enhances sampling diversity and, in turn, system performance.
>
> - **A baseline of DPO with Monte-Carlo sampling.** We would like to thank the reviewer for suggesting this experiment. This new ablation study, as shown above, shows that MC sampling underperforms DBS even with more samples. We start the sample size with $k=2$, as a single sample $k=1$ would have the probability terms $p_{\theta_i}(y^\alpha_i\mid \texttt{Pa}(y^\alpha_i))$ canceled out in the SysDPO-Sampling (Eq. 9 in Appendix B).
>
> We would like to once again thank the reviewer for their constructive comments and for acknowledging the importance of aligning compound AI systems. We hope these clarifications and additions will strengthen the paper and more clearly convey its contributions to the study of alignment in compound AI systems.

---

> > ### Comment · Reviewer_yR8i · 2025-08-03
> >
> > I appreciate the authors for conducting the additional experiments. They indeed help me gain more confidence in my assessment.
> >
> > > The resulting algorithm being conceptually simple is a result of its generality: it can be applied to any compound AI systems with a DAG factorization. Although the general scheme is simple, our implementations involve non-trivial procedures such as a tailored derivation of the tractable loss function for the LLM+Diffusion model system (Appendix D), and diverse sampling strategies used in both SysDPO-direct (Appendix F) and SysDPO-Sampling (Appendix B). Without such diversity-encouraged sampling procedures, MC sampling yields sub-optimal performance, as shown next.
> >
> > That is true. However, it also seems that the difficulty of implementing an effective approach for a specific task is left to aspects not addressed by the proposed framework itself. It would be helpful if the authors provided practical guidelines on implementation details to better demonstrate the usefulness of the proposed "framework".

---

> > > ### Author Response · Authors · 2025-08-04
> > >
> > > We thank the reviewer for the thoughtful and constructive suggestion. Providing practical guidelines would help clarify how to apply the SysDPO framework for specific systems. Key steps are summarized below:
> > >
> > > - **Factorize the system into a DAG.** Each node corresponds to a stochastic component (e.g., a call to a generative model). Edges indicate the flow of information (e.g., text).
> > > - **Attach log-probability evaluators to each node.** *LLMs*: use the model’s tokenizer-level log-likelihood. *Diffusion models*: compute a tractable surrogate (Appendix D).
> > > - **Sample diverse intermediate candidates.** Choose modality-appropriate diversity sampling (e.g., Diverse Beam Search for text).
> > > - **Jointly update parameters.** Accumulate SysDPO loss gradients across all trainable components and take an optimizer step.
> > >
> > > We will include a practical guideline and code release in the camera-ready version to better illustrate how SysDPO can be applied in practice. We appreciate the reviewer’s suggestion and hope this addition better clarifies the utility of the SysDPO framework.

---

> > > > ### Comment · Reviewer_yR8i · 2025-08-04
> > > >
> > > > Thank you for the response. I will keep the current (positive) score. Based on the discussion, I am more confident that the proposed approach is useful. The reason for the current score (not a higher score) is that the experiments are limited in showing the goodness of the proposed approach as a general framework. Still, I think that it will be a good baseline for future research in this direction.

---

### Official Review · Reviewer_sSJH · 2025-07-02

**Clarity:** 3
**Significance:** 3
**Originality:** 3
**Rating:** 5
**Confidence:** 4

**Summary:**

This paper is concerned with the alignment of compound AI systems, which consist of multiple interacting components such as large language models and image generators. Existing alignment methods, such as DPO or RLHF, are not directly applicable to such systems since they involve non-differentiable operations and the preferences cannot be directly related to each of the subcomponents making up the system. The authors model these systems as Directed Acyclic Graphs (DAGs) and propose two system-level alignment techniques based on DPO, called SysDPO-Direct and SysDPO-Sample. The first method requires a preference dataset where intermediate outputs for each system component are available, while this is not required for the second, approximate, method. The proposed methods are evaluated on two tasks and compared against several well-chosen baselines.

**Questions:**

_Q1_:
Are there insights into the computational cost of applying the SysDPO-Sample alignment technique?  Including a graph that shows the downstream task performance against the number of intermediate outputs sampled would be very insightful. Ideally, it would be good to report both the SysDPO-Sample and SysDPO-Direct technique for at least one of the applications, such that these methods can also be compared against each other.

_Q2_:
One aspect of the paper that was not entirely clear to me, is the role of the parameter beta in (1) the beta-perfect alignment definition and (2) in the DPO loss function. Is this one parameter that occurs in both the theoretical alignment definition and the practical optimization objective? If this link can be made more explicit, that would increase the clarity of section 3. It would also be helpful to address the presentational issues previously addressed.

_Q3_:
In order to make stronger claims on the effectiveness of the methods, it would be helpful to report additional, more holistic, system-specific performance metrics for both tasks. Ideally, the methods would be evaluated using a broader set of metrics commonly used in alignment research (such as MixEval scores for the language components of the systems or equivalent task-specific metrics for the other modalities).

**Ethical Concerns:**

["NO or VERY MINOR ethics concerns only"]

**Final Justification:**

I aim to keep my score, which seems to be on the higher side among the other reviewers.  However, I think the paper brings novel and interesting contributions (albeit not unexpected in scope, given recent evolutions).  Also, the paper has sufficient technical depth and I think it has the potential to become an interesting reference work in a very popular research direction. Also, I appreciate the efforts the authors have done in order to meet the reviewers' concerns, including my own.  The rebuttal was precies and extensive, and I was happy with the additional insight.  Looking forward to the announced future work (like with more realistic dynamic graph structures, based my weakness on static DAGs), for which I agree it is out of scope for the current paper.

**Limitations:**

yes

**Quality:**

4

**Strengths And Weaknesses:**

## Strengths:

The proposed approach introduces a novel and well-motivated formulation for compound AI systems by modeling them as Directed Acyclic Graphs (DAGs).  This structure enables a compact factorization of the system’s output likelihood, making it possible to apply end-to-end, gradient-based alignment. By extending the established Direct Preference Optimization (DPO) framework to system-level alignment, the authors address a timely and practical problem in aligning multi-component AI systems.

Another strength of the paper is the diversity of the evaluated modular AI systems. The first is a vision-language pipeline composed of an LLM and an image generator with a tree-like DAG structure. The second is a textual system consisting of two collaborating LLMs with a branched DAG structure. On both tasks, the reported evaluation metrics show the value of joint alignment through the SysDPO framework, compared to no or isolated / individual alignment.

_The paper is well-written, the contributions are intuitive, well-defined, and well-described, supported by valuable empirical results, and I think they are an unsurprising but important evolution in the direction of AI agents of the future._

## Weaknesses:

One key weakness of the proposed methods lies in scalability and data efficiency. The SysDPO-Direct variant requires system-specific preference data that includes not only final system outputs, but also intermediate outputs for each component. In practice, collecting such detailed annotations is likely to be difficult, especially for large or deeply modular systems with many interacting parts.

The alternative, SysDPO-Sample, avoids this requirement. However, it relies on sampling intermediate outputs to approximate the likelihood of the generated final outputs, which is computationally expensive. Accurately estimating the summation in Equation (5) requires these samples to cover a large, potentially infinite, probability space (the space of all possible intermediate outputs).  In the experiments, only two intermediate samples were drawn, which seems insufficient to reliably approximate this likelihood. This raises concerns about the quality and stability of the approximation, especially for complex systems with many intermediate outputs (potentially in different modalities).

A limitation of the proposed methods is the restricted applicability to systems that can be expressed as static DAGs. As acknowledged by the authors, many real-world modular AI systems cannot be easily or meaningfully reduced to a fixed DAG structure. For example, in the LLM-Diffusion experiment, the DAG is only valid because the number of generated images is fixed at three. If this number was input-dependent (as in many open-ended generation tasks), the structure of the DAG is no longer static and it is unclear if the alignment techniques can still be applied. More generally, many compound AI systems have dynamic workflows with cycles, feedback loops, etc. It would be meaningful to explore how SysDPO can be extended to such scenarios or discuss more extensively how the method can be adapted in future work to handle dynamic system behavior.

As indicated in the strengths section, the diversity of the evaluated modular AI systems is solid and the choice of baselines is sensible. However, the empirical evaluation can be elaborated. First, the SysDPO-Sample method is only evaluated on a single task and dataset and the evaluation set consists of only 193 data points (from the 12,900 available data points). Secondly, only two evaluation metrics are reported based on binary preference judgements. It would be meaningful to include additional metrics that are tailored to the evaluated system.

There are some presentational issues that could be improved. The formalization of compound AI systems as DAGs would benefit from more precision, for example, the meaning of edges is left vague (it is defined as the flow of information), the sets I and J (for the intermediate and final outputs) are not introduced, and it's ambiguous whether each parameter set (the thetas) is associated with a specific node (as presented in the text) or with an entire model component (as in the figure). Also, the motivating example in Figure 1 is not particularly compelling, as the difference between panels (a) and (b) is limited and subjective, making the intended contrast unclear. Additionally, as a minor remark, in Section 5.1, the same work is cited inconsistently as Zhang and Zhuang, which should be corrected.

---

> ### Author Rebuttal · Authors · 2025-07-31
>
> We sincerely thank the reviewer for the constructive and encouraging feedback. We will incorporate clarifications and revisions, as well as new experiments, in the camera-ready version. Details follow.
>
> - **Scalability, efficiency, and computational cost.**
>     - We thank the reviewer for highlighting efficiency and scalability. **This is indeed a challenge in our method and also a fundamental challenge of compound AI system alignment.** The fundamental challenge is due to how the preference signal is collected: the user preference is only given to the final output $z$ but not any intermediate outputs $y$.  In other words, it would be much easier if we had a preference oracle to compare two intermediate outputs directly, i.e., $\texttt{pref}(y^1\succ y^2\mid x)$, from which running standard DPO separately on each model would suffice. Unfortunately, this is ill-defined because the evaluation of a component in a compound system must depend on other components, i.e., there is no ground-truth preference oracle of the intermediate outputs that stands independent of the generative system itself.  Therefore, as you rightly point out, intermediate samples have to be drawn online constantly (SysDPO-Sampling) or have sufficient coverage (SysDPO-Direct). **The challenge is thus fundamentally a** **credit assignment problem under non‑differentiable, hidden intermediate interactions**, **and the fact that humans only have access to and give preferences to the final outputs.**
>     - **How we tackle the efficiency challenge.** The core idea of improving the sample efficiency of the intermediate outputs is diversity. For SysDPO-Direct, we adopt a set of diverse styles of prompting to construct the preference dataset (detailed in Appendix F). For SysDPO-Sampling, we adopt diverse beam search (DBS) to improve the diversity of the intermediate outputs.
>         - **We run a new ablation study to show that DBS outperforms Monte Carlo sampling with the same sample budget, highlighting the importance of diversity for efficiency.** We begin by comparing Diverse Beam Search (DBS) with Monte‑Carlo (MC) sampling and then examine how performance scales with the number of intermediate candidates sampled during training. Our experiments reveal two clear trends: (i) within the tested range, increasing the sampling budget $k$ has little to no impact on overall performance; and (ii) DBS achieves higher win‑rates than MC with the same sample count.
>         - **Experimental Settings and sample details**. Building on the LLM collaboration system in section 5.2, each training instance starts from input $x$, model $\psi_1$, and $\psi_2$. During training, we sample $k$ intermediate candidates $y^1, \dots y^k \sim p_{\psi_1}(\cdot | x)$ and leverage them to calculate the loss that jointly optimize $\psi_1, \psi_2$. At evaluation time, we eliminate sampling stochasticity by setting the temperature to 0 and deterministically selecting $y^\* = \max_y p_{\psi_1}(y | x)$  and $z^\* = \max_z p_{\psi_2}(z | x)$. We evaluate the system performance based on the quality of $z^\*$.
>         - **Sampling methods.** For DBS, We obtain $k$ candidates $y^1, \dots, y^k$ using $k$ beam groups and a diversity penalty of 20; we report results for $k=2,4$. For MC sampling, we independently sample $y^1, \dots, y^k$ with temperature 1.0 under different random seeds, evaluating $k=\{2,3,4,5\}$.
>         - **Results.** The table below summarizes the win‑rate of each configuration. **DBS outperforms MC with the same number of sampled candidates.** Both the methods are consistent with varying $k$ within the test range, **suggesting that 2 samples may be sufficient without drastically increasing the budget.** Compared to DBS, MC often draws candidates that are near‑duplicates.
>
>         | Method | k | WR-Prompted (%) |
>         | --- | --- | --- |
>         | DBS | 2 | 68.5 |
>         | DBS | 4 | 68.2 |
>         | MC | 2 | 66.8 |
>         | MC | 3 | 67.0 |
>         | MC | 4 | 66.7 |
>         | MC | 5 | 66.0 |
>     - **Computational cost.**  Aligning the system of two LLMs jointly took approximately 30 hours on a single NVIDIA H200 GPU. We acknowledge that broader scaling is out of our limited academic budget and remains open. We therefore see this work as a principled first step that lays the groundwork for future large-scale investigations.
> - **Fixed DAG structure.**
>     - We appreciate the reviewer’s insight and agree that many real-world systems feature dynamic, input-dependent control flow.  **For settings with loops or dynamic branches, we envision two extensions**. (i) **time-unrolling for recurrent interactions.** For systems with loops, one can unroll the interaction over time, yielding a finite-horizon DAG, similar to back-propagation through time. (ii) **stochastic masking.** Some nodes may learn a policy over which downstream modules to invoke, where the random variable is a mask over the node’s out edges. Its probability naturally integrates into the SysDPO framework. Exploring these directions is an exciting future work.
> - **The role of $\beta$ in $\beta$-perfect alignment.**
>     - **It is very interesting how $\beta$-perfect alignment can be precisely interpreted from many different perspectives.** However, we acknowledge that the explanation of the original manuscript can be improved for better clarity. Below, we first give the intuition of $\beta$-perfect alignment. Then, we discuss how the $\beta$ is the **same** as the $\beta$ appearing in the DPO/RLHF objective as the KL divergence.
>         - **$\beta$-perfect alignment of generative models.** Given an input $x$, consider two versions of the generated output $z^1$and $z^2$. Suppose there is a ground truth preference oracle $\texttt{pref}(z^1\succ z^2\mid x)\in (0, 1)$, which states how much  $z^1$ is preferred over $z^2$.  The model $\theta^\*$ is $\beta$-perfectly aligned to the preference oracle $\texttt{pref}(\cdot)$ if for any $x\in\mathcal{X}$ and $z^1, z^2 \in \mathcal{Z}$ such that $$\texttt{pref}(z^1\succ z^2\mid x) = \frac{p_{\theta^\*}(z^1\mid x)^\beta}{p_{\theta^\*}(z^1\mid x)^\beta+p_{\theta^\*}(z^2\mid x)^\beta} \qquad\qquad{(\text{Definition 1})}$$
>         - **What varying $\beta$ does**. We can understand the effect of varying $\beta$ through a concrete example. Suppose we have a preference oracle of $\texttt{pref}(z^1\succ z^2\mid x)=0.8$ on data $(x, z^1, z^2)$. Then, we obtain a $\beta$-perfectly aligned model $\theta^\*$ that satisfies the alignment equation (Definition 1). We may rearrange the alignment equation (Definition 1) to be
>         $$
>     \texttt{pref}(z^1\succ z^2\mid x) = \frac{1}{1+\left(\tfrac{p_{\theta^\*}(z^2\mid x)}{p_{\theta^\*}(z^1\mid x)}\right)^{\beta}}
>     $$ If $\beta\ll 1$ is close to $0$, it would require the ratio $p_{\theta^\*}(z^2\mid x)/p_{\theta^\*}(z^1\mid x)\ll 1$ for the equation to hold: the aligned model almost always output the most preferred result deterministically. Conversely, if $\beta \gg 1$, it would require the ration  $p_{\theta^\*}(z^2\mid x)/p_{\theta^\*}(z^1\mid x)\approx 1$, i.e., approaching a uniform policy. **Therefore, in this view, $\beta$ can be interpreted as temperature**.
>
>         - **The perspective of DPO/RLHF**. As shown in Proposition 1, the optimal solution $\theta^\*$ to the DPO or the RLHF objective function, when the reference model follows a uniform distribution, achieves $\beta$-perfect alignment. Most interestingly, the $\beta$ is **precisely** the KL regularization strength parameter that appears in both the DPO and RLHF objective. **Thus, in this view, $\beta$ can be interpreted as the strength of KL regularization.**
> - Suggested clarification presentation regarding formulating compound AI systems as DAGs, Figure 1 description, and typos will be incorporated in the camera-ready version.
>
> We thank the reviewer once again for the encouraging feedback and for acknowledging the importance of compound AI systems alignment. We hope these clarifications and additions will strengthen the paper and more clearly convey its contributions to the study of alignment in compound AI systems.

---

> > ### Comment · Reviewer_sSJH · 2025-08-06
> >
> > The concerns raised in my review have been substantially addressed in the rebuttal, particularly through the addition of new experiments comparing Diverse Beam Search (DBS) and Monte Carlo sampling, and through the clarification of the theoretical framing.
> > That being said, I went through the reviews, and I agree with Reviewer 7XLb that the approximation of the intractable marginal likelihood (equation 5) using DBS would benefit from a deeper discussion regarding its theoretical soundness (in finite samples). In addition, the example in figure 1 appears to have caused confusion among multiple reviewers.  Also, the intended contrast between the two image sequences is not very clear, which hinders the motivation of the paper.
> > Still, for myself I do think this paper is worth publication at NeurIPS, and a logical step in the series of recent alignment papers.

---

> > > ### Author Response · Authors · 2025-08-06
> > >
> > > We thank the reviewer for the supportive assessment and for recognizing that the main concerns have been substantially addressed. Based on your feedback, we will make the following updates in the camera-ready version: (i) while a rigorous finite-sample theory is indeed non-trivial and beyond the current scope, we will include a discussion on the theoretical soundness of the DBS approximation in finite-sample settings through the lens of coreset theory (as mentioned in our reply to Reviewer 7XLb); (ii) revise Figure 1 and its caption to emphasize the intended contrast (the progression of anger) and to avoid potential misunderstandings. We appreciate your insights, which will help further strengthen the final version of the paper.

---

> > > > ### Comment · Reviewer_sSJH · 2025-08-08
> > > >
> > > > I again went over my review and the rebuttal, as well as the rebuttals to the other reviews.  I aim to keep my score, which seems to be on the higher side, since I think the paper brings novel and interesting contributions (albeit not unexpected in scope, given recent evolutions), and IMHO is worthy of NeurIPS in terms of technical quality and potential impact.  Also, I appreciate the efforts the authors have done in order to meet the reviewers' concerns, including my own.

---

### Official Review · Reviewer_7XLb · 2025-07-02

**Clarity:** 2
**Significance:** 2
**Originality:** 2
**Rating:** 3
**Confidence:** 3

**Summary:**

This paper highlights the difficulty of fine-tuning models from preferences when those models are used as modules within a multi-model system. The two running examples are

1. An image-generation system which takes a user prompt, uses an LLM to generate input prompts to an image diffusion model, then produces a sequence of images.
2. A multi-step LLM system, which uses one LLM forward pass to improve a prompt, then another to respond to the user using that improved prompt.

Because these handoff points between models are in text, they are not differentiable, so we cannot simply pass gradients through the whole system. Instead, the authors propose to model the system as a Bayesian network where each node is a model (implicitly parameterizing a likelihood function) and the edges map the data flow. This admits a factorization of the overall system's likelihood function which can be used to construct a differentiable loss function. If we have a preference dataset which includes the intermediate models' outputs, we can plug that directly into the DPO loss function; the authors term this algorithm SysDPO-Direct. If we do not have the intermediate outputs, the true likelihood requires marginalizing over them, which is intractable, and the authors propose to approximate the likelihood with samples generated by Diverse Beam Search rather than Monte Carlo approximation, terming this algorithm SysDPO-Sampling.

In their theoretical analysis, the authors define $\beta$-Perfect Alignment. This $\beta$ is a temperature parameter, the inverse of the *inverse*-temperature parameter $\beta$ from the Boltzmann rational model. My understanding is that if there is an underlying utility function (which I think is an implicit assumption underlying the existence of this sort of preference oracle), then a probabilistic generative model is $\beta$-perfectly aligned if and only if its outputs match the choice probabilities of a Boltzmann rational agent with inverse temperature $\beta$ acting with that utility function. If so, this is equivalent to the classic RLHF objective with a rationality parameter (as described in e.g. [1]).

The paper then argues that both variants of SysDPO achieve perfect alignment in the infinite-data regime (SysDPO-Sampling is only addressed in the case where infinite samples can be drawn and thus the value of the intractable integral over unobserved variables is known).

The paper concludes with experiments showing SysDPO outperforming some baselines in the two settings described in the running examples. For the image generation setting, this entails a derivation of an optimizable likelihood for the diffusion model.

[1] Hong Jun Jeon, Smitha Milli, and Anca Dragan. 2020. Reward-rational (implicit) choice: A unifying formalism for reward learning. Advances in Neural Information Processing Systems, 33:4415–4426.

**Questions:**

1. Can you provide any more detailed theoretical analysis of SysDPO-Sampling in the finite sample case?
2. Can you clarify the statement "The fact that SysDPO-Direct requires Assumption 1 for perfect alignment, while standard DPO does not, highlights a fundamental challenge of compound AI system alignment." ? Since standard DPO doesn't have $\mathcal Y$, the parallel assumption would be that all outputs have some positive probability to be sampled from $\mathcal D$, which seems like a necessary condition for standard DPO to achieve perfect alignment. If I'm incorrect, please help me understand that and what fundamental challenge is highlighted here; the later sentences in that same paragraph seem unrelated to that statement.
3. Can you provide any more firm understanding of what $\beta$-perfect alignment actually means? The statement that jumps out at me is that it's a generalization of the IIA condition from Luce's choice axioms, to which it corresponds when $\beta = 1$. Is there some meaningful correspondence that you can speak to there, perhaps including what varying $\beta$ would mean under that interpretation?
4. Some additional experiments that would go a long way in convincing me:
  - A more general multi-image generation task (perhaps just open-ended preferences over sequences of images, or pairs of images)
  - Aligning multiple differently-structured LLM collaboration systems (i.e. different DAGs) on the same dataset for 5.2
  - A more significant hyperparameter study on the sampling in SysDPO-sampling
5. Please clarify the training setup of the Separate-DPO baseline for Experiment 5.2.

---

A couple of significant typos I noticed:
1. The text says the OC Ratio for SysDPO-Direct in 5.1 was 70%, whereas the table says 73%
2. In the third line (second implication) of the math in line 575 in the appendix, $z^w$ and $z^l$ are swapped.

---

All told, this is a great problem to be tackling with such a general-purpose framework, a gerat  and I look forward to further developments on  empirically validating that it can work at a practical scale.

**Ethical Concerns:**

["NO or VERY MINOR ethics concerns only"]

**Final Justification:**

The clarifications on $\beta$-perfect alignment are helpful.

On the finite-sample analysis, I remain hopeful for a stronger theoretical result, as much of the paper is devoted to theoretical analysis and finite-sample SysDPO-Sampling is the core setting and algorithm (since as you say the fundamental challenge is that humans only give preferences over the final outcomes, not the intermediate model outputs). Nonetheless your discussion here along with the new sample-size experiment give some foothold for future work to progress in that direction.

For practical applications, the new experiment with a multi-LLM system is a good start on proof-of-concept, and improves the paper.

I have increased my score to a 3. There is great work here, and while there are remaining gaps in what I would like to see in a NeurIPS publication on this topic, I fully respect if the other reviewers and ACs find it complete in its current form. Thank you.

**Limitations:**

yes

**Quality:**

3

**Strengths And Weaknesses:**

## Strengths

The framework is very general, and provides an outline for how to jointly optimize any static multi-model system from preferences so long as a likelihood function (or likelihood estimator) is available for each model, whether or not the preference data contains the outputs of the intermediate models.

In particular, proposing algorithms for the case without intermediate results provides an alignment framework that could work for multiple different systems designed for the same task. One could have input-output pairs for the same task, design several different LLM collaboration pipelines to tackle the task, and use SysDPO-Sampling to align all of them from that single dataset.

The proofs are rigorous (I read and understood all but the diffusion model derivations in Appendix D) and provide firm confidence in the statements that RLHF, DPO, and SysDPO-Direct achieve $\beta$-perfect alignment in the infinite-data regime. The related statement for SysDPO-Sampling is not as rigorously argued, but seems true in the infinite-data, infinite-samples regime.

The derivation of the tractable likelihood function for the diffusion model in Appendix D seems substantial, and I would consider promoting it more centrally as a contribution of the paper:
1. This specific derivation, if it is novel, could be relevant for other work that needs such a likelihood estimator;
2. This is an example demonstrating a more general point: that the Bayesian network factorization of the full system's likelihood function doesn't strictly require that we have a likelihood function for each model if we can derive a sufficiently good estimator.

The experiments seem decently well-precedented and provide some evidence that joint alignment outperforms marginally aligning individual components. They also provide some evidence that SysDPO-Sampling is effective in practice with small sample sizes.

## Weaknesses

I was confused at first by the example in Figure 1. After reading more of the paper I have come to understand the point being made: the second cat in (a) is not visibly more angry than the first cat in (a), whereas the cats in (b) show a more significant progression of anger; this highlights that the models' cooperation is inconsistent. My initial interpretation was that these were both clearly bad examples, because both progressions contain three clearly different cats. As a sanity check I showed the figure to a couple of family members (not in the AI field) and their interpretation was the same. This left me a little bit lost my first time reading the paper as to what practical problem we were grounding the discussion with.

Generally, the SysDPO-Direct idea feels like a very minor contribution theoretically. There is quite a bit of room to validate its effectiveness and to work through practical issues in scaling it to real practical problems (cf the original RLHF paper), but this work as written up here significantly emphasizes the theoretical aspects of this algorithm.

On the other hand, while Monte Carlo approximation of the sum when marginalizing over unobserved variables in the Bayesian network is a natural idea, the paper proposes to instead use Diverse Beam Search to generate samples, which is an interesting proposition with some intuitive motivation. On reading that I immediately wondered if that would lead to any bias in the likelihood estimation and whether that would matter in practice. For a paper that lends so much of its space in both main text and appendix to theoretical analysis and derivations, I was disappointed that for the one theoretical question that I found myself wanting an answer to was only addressed by pointing out that since the Diverse Beam Search samples are distinct, drawing infinitely many samples approaches identity and so guarantees perfect alignment. Yes, but the whole motivation is that the sum is intractable, and in the experiments we only draw a single sampled pair for SysDPO sampling. Is there anything we can say about the soundness of using DBS instead of MC in the finite data case?

I struggled to garner much intuitive meaning of $\beta$-perfect alignment. After reading the paragraph about interpreting that definition and Proposition 1, I arrived at the understanding I wrote in my summary above. But that's a bit circular with Proposition 1, and I find myself feeling that the $\beta$-perfect alignment definition does not contribute to my understanding at all. Rather, I end up at "an algorithm achieving perfect alignment means it gets the same result as RLHF/DPO".

Broadly speaking, I struggled to move my subjective belief significantly based on the results from the Experiments section. The first example feels a bit narrow and contrived -- a more specific task than one tends to run post-training DPO to align to. I would have liked to see something more general to motivate the LLM + image diffusion case. For the second example, I was a bit confused about the Separate-DPO baseline (the most relevant one to the core point of the paper). What data is being used to DPO-tune the two language models? Isn't an assumption here that we don't have factorized preference data that we could use to independently tune the models? I would have also liked to see much more exploration of the impact of the sample size $\alpha$ in SysDPO-Sampling. What about, rather than sampling four responses then choosing the most contrastive pair, just using more samples?

---

> ### Author Rebuttal · Authors · 2025-07-31
>
> We sincerely appreciate the reviewer’s insightful feedback and recognition of the compound AI systems alignment problem. To our knowledge, SysDPO is the first principled framework that jointly aligns the parameters of an entire compound AI system modelled as a DAG. For better fluency, we address the reviewer's comments and questions in the order of their appearance in the manuscript.
>
> - **Detailed explanation of $\beta$‑perfect alignment:** We thank the reviewer for suggesting the perspective of the Boltzmann rational model. **It is very interesting how $\beta$-perfect alignment can be precisely interpreted from many different perspectives.** Details follow.
>     - $\beta$**-perfect alignment of generative models.** Given an input $x$, consider two versions of the generated output $z^1$ and $z^2$. Consider a ground truth preference oracle $\texttt{pref}(z^1\succ z^2\mid x)\in (0, 1)$, which states how much  $z^1$ is preferred over $z^2$.  The model $\theta^\*$ is $\beta$-perfectly aligned to the preference oracle $\texttt{pref}(\cdot)$ if for any $x\in\mathcal{X}$ and $z^1, z^2 \in \mathcal{Z}$ such that $$
> \texttt{pref} (z^1 \succ z^2 \mid x) = \frac{p _ {\theta^\*} (z^1\mid x)^\beta}{p_{\theta^\*}(z^1\mid x)^\beta+p_{\theta^\*}(z^2\mid x)^\beta} \qquad\qquad{(\text{Definition 1})}
> $$
>
>     -  **What varying $\beta$ does**. Suppose we have $\texttt{pref}(z^1\succ z^2\mid x)=0.8$ on data $(x, z^1, z^2)$. Then, given a $\beta$-perfectly aligned model $\theta^\*$ that satisfies the alignment equation (Definition 1). Rearranging the alignment equation: $$
>     \texttt{pref}(z^1\succ z^2\mid x) = \frac{1}{1+\left(\tfrac{p_{\theta^\*}(z^2\mid x)}{p_{\theta^\*}(z^1\mid x)}\right)^{\beta}}
>     $$ If $\beta\ll 1$ is close to $0$, it would require the ratio $p_{\theta^\*}(z^2\mid x)/p_{\theta^\*}(z^1\mid x)\ll 1$ for the equation to hold: the aligned model almost always output the most preferred result deterministically. Conversely, if $\beta \gg 1$, it would require the ration  $p_{\theta^\*}(z^2\mid x)/p_{\theta^\*}(z^1\mid x)\approx 1$, i.e., approaching a uniform policy. **Therefore, in this view, $\beta$ can be interpreted as temperature**.
>
>     - **The perspective of the Boltzmann rational model**. With our notations, the generative policy $p_{\theta^\*}(\cdot\mid x)$ of $z$ is Boltzmann rational if $p_{\theta^\*}(z\mid x)\propto \exp(r^\*(x, z)/\beta)$ where $r^\*(x, z)\in \mathbb{R}$ is a ground-truth reward function of the input-output pair. Note that the $\beta$ here is inverted compared to conventional Boltzmann rational literature, as the reviewer rightly point out. We can verify that this Boltzmann rational policy is indeed $\beta$-perfectly aligned: $$
>     \frac{p_{\theta^\*}(z^1\mid x)^\beta}{p_{\theta^\*}(z^1\mid x)^\beta+p_{\theta^\*}(z^2\mid x)^\beta}=\frac{\exp(r^\*(x, z^1))}{\exp(r^\*(x, z^1))+\exp(r^\*(x, z^2))}=\texttt{pref}(z^1\succ z^2\mid x)
>     $$ where the last equation is the definition of the Bradly-Terry preference model (Eq. 1 in the main text). **Therefore, $\beta$ can be interpreted as the inverse of rationality.**
>
>     - **The perspective of DPO/RLHF**. As shown in Proposition 1, **the $\beta$ is precisely the KL regularization strength parameter** appears in both the DPO and RLHF objective.
> - **Clarification on the fundamental challenge of aligning compound AI systems compared to standard single-model alignment**
>     - The fundamental challenge is due to how the preference signal is collected: the user preference is only given to the final output $z$ but not any intermediate outputs $y$. Since the evaluation of a component in a compound system must depend on other components, there is no preference oracle to compare two intermediate outputs directly, i.e., $\texttt{pref}(y^1\succ y^2\mid x)$. Therefore, intermediate samples have to be drawn online constantly (SysDPO-Sampling) or have full coverage (SysDPO-Direct). The fundamental challenge is thus **credit assignment under non‑differentiable, hidden intermediate interactions**, **and the fact that humans only have access and give preferences to the final outputs.**
> - **Finite‑sample analysis for SysDPO‑Sampling with Diverse Beam Search (DBS) and MC Sampling**
>     - Consider a two-generative-model collaboration system (Figure 2 (b)). As shown in Section 2.3, SysDPO-Sampling aims to approximate the intractable generation probability $p_{\theta}(z\mid x)=\sum_{y\in \mathcal Y} p_{\theta_2}(z\mid y, x)p_{\theta_1}(y\mid x)$ with finite samples $\widehat{\mathcal{Y}}:=\\{ y^\alpha \\} _ \alpha$ such that approximately  $p_{\theta}(z\mid x)\propto\sum_{\alpha} p_{\theta_2}(z\mid y^\alpha, x)p_{\theta_1}(y^\alpha\mid x)$. **This can be viewed as a coreset selection problem.** I.e., the goal is to find a diverse subset $\widehat{\mathcal Y}$ of representative points to approximate the full set $\mathcal Y$. Denoting $\mu_{\mathcal Y}:=\tfrac{1}{|\mathcal Y|}\sum_{y\in \mathcal Y} p_{\theta_2}(\cdot\mid y, x)p_{\theta_1}(y\mid x)$ (similarly for $\mu_{\widehat{\mathcal Y}}$ ), we aim to minimize the approximation error $\epsilon:=\|\mu_{\mathcal Y}-\mu_{\widehat{\mathcal Y}}\|$. **Therefore, a strategically selected subset often outperform random MC sampled subset, where the strategy typically involves selecting representative and diverse points [A, B, C].** DBS can be seen as a greedy coreset‑selection algorithm where, at every decoding step, it keeps the subset that jointly maximizes a combined objective of generation likelihood and diversity.
>     - **Toward a finite-sample analysis.** The core of finite-sample analysis is the rate of approximation error $\epsilon$ w.r.t. the budget $k:=|\widehat{\mathcal{Y}}|$. Although it is unclear what the exact error $\epsilon$ of DBS is, coreset algorithms generally achieve a worst-case guarantee of $\epsilon=\mathcal O(\tfrac{1}{\sqrt{k}})$ [A]. For example, consider finite sample space $\mathcal Z$, and denote the sampled probability mass function as $v_y:=p_{\theta_2}(\cdot\mid y, x)p_{\theta_1}(y\mid x)\in \mathbb{R}^{|\mathcal Z|}$. Assume all $v_y$ is $\ell_2$-bounded by $D$. The approximate carathéodory theory [D] states that we can always find a subset $\widehat{\mathcal{Y}}\subset \mathcal Y$ of size $k$ such that the approximation error  $\epsilon=\mathcal O(\tfrac{D}{\sqrt{k}})$. To make the bound tighter, we need fine-grained analysis that exploits the concrete structure of the distribution, which is an exciting but non-trivial future work. Nonetheless, in practice, coreset can be significantly better as shown in prior works [A, B, C], as well as our experiments shown next.
>     - **Experimental Settings**. In this newly conducted experiment, we build on the LLM collaboration system in Section 5.2. During training, we sample $k$ intermediate candidates $y^1, \dots y^k \sim p_{\psi_1}(\cdot | x)$ and use them to calculate the loss. At evaluation time, we eliminate sampling stochasticity by setting the temperature to 0 and deterministically selecting $y^\* = \max_y p_{\psi_1}(y | x)$  and $z^\* = \max_z p_{\psi_2}(z | x)$. We evaluate the system performance based on the quality of $z^\*$.
>     - **Sampling methods.** For DBS, We obtain $k$ candidates using $k$ beam groups and a diversity penalty of 20; we report results for $k=2,4$. For MC sampling, we independently sample $k$ candidates with temperature 1.0 under different random seeds, evaluating $k\in\{2,3,4,5\}$.
>     - **Results.** The table below shows the win‑rate of each configuration. **DBS outperforms MC with fewer number of sampled candidates**. Both methods are consistent with varying $k$ within the test range, **suggesting that 2 samples may be sufficient without drastically increasing the budget**. Compared to DBS, MC often draws candidates that are near‑duplicates.
>
>
>         | Method | k | WR-Prompted (%) |
>         | --- | --- | --- |
>         | DBS | 2 | 68.5 |
>         | DBS | 4 | 68.2 |
>         | MC | 2 | 66.8 |
>         | MC | 3 | 67.0 |
>         | MC | 4 | 66.7 |
>         | MC | 5 | 66.0 |
> - **Experiments on aligning differently-structured LLM collaboration systems.**
>     - **We include a new experiment on a three‑LLM system:** LLM 1 and LLM 2 independently generate responses to the input question, and LLM 3 synthesizes these responses into a final answer. This setup extends our two-LLM system to work with more components. We train this system jointly with SysDPO and evaluate it using the WR‑Prompted metric. Without changing the hyperparameter, the aligned system achieves a 58 % win rate against its pre‑alignment version, showing promising potential for scalability. While broader experiments with many components are beyond our limited academic budget, we believe this result demonstrates a solid foundation for future scaling.
> - **Due to the space limitation, we summarize the remaining clarifications below**. Luce’s choice axioms introduce a probabilistic framework for modeling how people make choices. As we view the generative model $p_{\theta^\*}(z\mid x)$ as a choice model, the axioms (Lemma 3 in Luce et al.) coincide with our Definition 1 with $\beta=1$. In Experiment 5.2, the Separate-DPO baseline is done by training each model on the same task‑level dataset (Orca‑DPO‑Pairs) independently. In addition, clarification about Figure 1 and suggested typos will be incorporated in the camera-ready version.
>
> We thank the reviewer again for engaging with our work. We hope these clarifications and additions will strengthen the paper and more clearly convey its contributions to the study of compound AI systems alignment.
>
> **References**
>
> [A] Campbell et al. Automated Scalable Bayesian Inference via Hilbert Coresets.
>
> [B] Zhang et al. Bayesian Coresets: Revisiting the Nonconvex Optimization Perspective.
>
> [C] Sener et al. Active Learning for Convolutional Neural Networks: A Core-Set Approach
>
> [D] Mirrokni et al. Tight Bounds for Approximate Carathéodory and Beyond
>
>  Luce et al. Individual choice behavior.

---

> > ### Author Response · Authors · 2025-08-05
> >
> > We thank the reviewer again for engaging with our paper. As the discussion period is nearing its end, we would like to kindly follow up in case you have any follow-up questions. We would be happy to clarify or elaborate on any points that may assist in your final assessment.

---

> > > ### Comment · Reviewer_7XLb · 2025-08-08
> > >
> > > Thank you for the detailed and thoughtful responses to my review. I would like to apologize for my late response; I have been down with Covid the past week and unable to engage deeply.
> > >
> > > The clarifications on $\beta$-perfect alignment are helpful.
> > >
> > > On the finite-sample analysis, I remain hopeful for a stronger theoretical result, as much of the paper is devoted to theoretical analysis and finite-sample SysDPO-Sampling is the core setting and algorithm (since as you say the fundamental challenge is that humans only give preferences over the final outcomes, not the intermediate model outputs). Nonetheless your discussion here along with the new sample-size experiment give some foothold for future work to progress in that direction.
> > >
> > > For practical applications, the new experiment with a multi-LLM system is a good start on proof-of-concept, and improves the paper.
> > >
> > > I have increased my score to a 3. There is great work here, and while there are remaining gaps in what I would like to see in a NeurIPS publication on this topic, I fully respect if the other reviewers and ACs find it complete in its current form. Thank you.

---

### Official Review · Reviewer_jnDe · 2025-07-23

**Clarity:** 3
**Significance:** 3
**Originality:** 2
**Rating:** 4
**Confidence:** 3

**Summary:**

The paper studies alignment to human preferences in compound AI systems, such as those with multiple interacting components such as LLMs and tools. The authors formulate the compound AI system as a DAG to model component interactions and data flow and introduce SysDPO. They conduct empirical evaluations on two settings: joint alignment of language and diffusion models as well as LLM collaboration.

**Questions:**

Please see weaknesses

**Ethical Concerns:**

["NO or VERY MINOR ethics concerns only"]

**Limitations:**

Yes

**Quality:**

2

**Strengths And Weaknesses:**

Strengths:
- The paper studies *joint* preference optimization in compound AI systems, which is an important problem (e.g., applications in multi-agent LLMs, LLMs+RAGs, multi-LLM routing systems, as the authors explain).
- The problem considered in the paper is nontrivial as it non-differentiable interactions between components, difficultly with the assignment of credit and preferences across components, and decomposing the system-level preferences.
- Formulating the compound AI system as a DAG and deriving a preference optimization algorithm based on the factorized probabilities is interesting.
- Experiments show SysDPO performs better compared to prompted system, BoN sampling, and separate DPO, showing the necessity of joint alignment.

Weaknesses:
- Theoretical analysis in Section 3.1 include simple extensions of the properties of DPO and does not offer new insights.
- Paragraph below Theorem 1 states that Assumption 1 is required; however, the theorem seems to only show that the statement holds under Assumption 1 and not the necessity of this assumption.
- Experiments are limited to two component compound systems, and it’s unclear how the performance of joint PO compares with separate PO.

---

> ### Author Rebuttal · Authors · 2025-07-31
>
> We sincerely thank the reviewer for the constructive and encouraging feedback. We will incorporate clarifications and revisions, as well as new experiments in the camera-ready version. Details regarding your comments follow.
>
> - **Insights of the theoretical analysis in Section 3.1 and how it extends standard DPO**
>     - Section 3.1 indeed revisits standard DPO and RLHF to show that they achieve $\beta$-perfect alignment. The purpose is to justify the definition of $\beta$-perfect alignment and establish a connection with existing results. Our theoretical contribution and new insights are discussed in Section 3.2: by factoring a directed acyclic graph (DAG) of components, we prove that **SysDPO achieves the same β‑perfect alignment guarantee at the system level (under Assumption 1), even when the system has a non‑differentiable structure. To our knowledge, this is the first formal alignment guarantee in such compound AI system alignment.**
>     - Moreover, in the LLM+Diffusion model setting, deriving a tractable SysDPO loss function is non-trivial: diffusion models, unlike LLMs, do not directly output the log-likelihood of generated outputs. We address this through a tailored derivation (shown in Appendix D), extending the single-model Diffusion DPO (Wallace et al.). **This is an important and nontrivial technical contribution not emphasized in the main text due to space limitations**.
>     - We will highlight these distinctions in the camera‑ready and move the key part of Appendix D into the main text.
> - **Theorem 1 seems to only show that the statement holds under Assumption 1 and not the necessity of this assumption.**
>     - We thank the reviewer for catching this. Yes, Assumption 1 is sufficient for proving $\beta$-perfect alignment, but it may not be necessary. In practice, diverse samples often suffice. We will clarify this point and revise the statement in the camera-ready version.
> - **Experiments beyond two components.**
>     - We acknowledge the importance of evaluation on large-scale multi-component systems. **We include a new experiment on a three‑LLM system:** LLM 1 and LLM 2 independently generate responses to the input question, and LLM 3 synthesizes these responses into a final answer. This setup extends our two-LLM system to work with more components. We train this system jointly with SysDPO and evaluate it using the WR‑Prompted metric. Without changing the hyperparameter, the aligned system achieved a 58 % win rate against its pre‑alignment version, showing promising potential for scalability. While broader experiments with many components are beyond our limited academic budget, we believe this result demonstrates that our framework provides a solid foundation for future scaling.
> - **Joint policy optimization vs separate policy optimization.**
>     - Thank you for raising this important distinction. **Even in the two-component setting, we observe that separate DPO underperforms joint SysDPO** as shown in Table 2. In this LLM+LLM setting, LLM 1 drafts an initial response, and LLM 2 refines it. For Separate-DPO, we train each model on the same task‑level dataset (Orca‑DPO‑Pairs) independently.
>     In contrast, SysDPO aligns both models jointly, allowing updates that reflect their collaboration dynamics. This leads to a more coordinated and effective system. We will highlight this difference more explicitly in the camera-ready version, as it demonstrates the necessity of joint alignment.
>
> We would like to once again thank the reviewer for their constructive comments and for acknowledging the importance of aligning compound AI systems. The clarifications provided above will strengthen the paper both theoretically and empirically.
>
> **References.**
> Wallace et al. Diffusion Model Alignment Using Direct Preference Optimization.

---

### Note · Authors · 2025-08-12

We thank the reviewers for their constructive and valuable feedback. Below we restate the scope and key contributions, then summarize the key updates made following the reviewers' suggestions.

## Contributions and scope

This work addresses **system-level alignment of compound AI systems**, where fundamental challenges involve credit assignment under non‑differentiable and hidden intermediate interactions among components.

Concretely, in this work we:

1. **Formalize** the system-level preference alignment problem of compound AI systems.
2. **Propose** a principled joint optimization objective (SysDPO) and two practical training variants (SysDPO-Direct and SysDPO-Sampling).
3. **Demonstrate** that SysDPO, as a single elegant framework, can be applied to aligning various structured compound AI systems such as LLM+Diffusion and LLM+LLM collaborative systems.

To our knowledge, this is the **first work to explicitly formulate the alignment problem for compound AI systems and offer a general, extensible solution.**

## Key Updates Post-Rebuttal

**Clarification and analysis on intermediate samples (SysDPO-Sampling)**. We clarified the sampling procedure in SysDPO-Sampling with empirical ablations theoretical discussion. For each input to the system we draw $k$ intermediate samples using either Monte Carlo (MC) or Diverse Beam Search (DBS). The results show that diversity is more important than quantity: DBS with $k=2$ provides high-quality candidates and outperforms MC. We also discuss a coreset-style analysis that justifies why DBS outperforms MC for SysDPO-Sampling.

**Beyond two components.** We included a three-LLM system where two models answer independently and a third synthesizes their outputs. Results show that joint SysDPO improves win rates over the unaligned baseline, suggesting the method naturally extends to richer collaboration systems.

**Clarification of $\beta$-perfect alignment and Figure 1.** We expanded the explanation of the definition of $\beta$-perfect alignment, showing it admits multiple equivalent interpretations (temperature scaling, Boltzmann rationality, KL regularization). Figure 1 was revised to highlight the intended conceptual contrast and prevent misinterpretation.

We believe these clarifications and additional results further strengthen the paper, providing more evidence for **SysDPO as a general solution for aligning compound AI systems, an increasingly important challenge for the field.**

---

### Decision · Program_Chairs · 2025-09-17

**Decision:**

Accept (poster)

**Comment:**

The paper studies how workflows consisting of multiple non-differentiable components can be jointly optimized using datasets of preferences over the end outputs of the workflow. The paper views the workflow as a Bayesian graphical model and derives likelihood estimators for each component, which allows them to define a differentiable loss function over the entire workflow and optimize via a variant of DPO.

All the reviewers agreed that the problem is well-motivated and significant, and the proposed approach is novel. The reviewers raised several questions and gave constructive suggestions to improve the experimental evidence; the authors incorporated the feedback, especially the additional experiments comparing to Monte Carlo sampling, evaluating on a larger (3-component) workflow, and clarifications on the derivations substantially strengthened the paper. No reviewer objects to the paper being accepted when revised with the additional experiments and discussions.